



# Periodic input of dust over the Eastern Carpathians during the Holocene linked with Saharan desertification and human impact

Jack Longman[1], Daniel Veres[2], Vasile Ersek[1], Ulrich Salzmann[1], Katalin Hubay[3], Marc Bormann[4], Volker Wennrich[5], Frank Schäbitz[4]

[1] Department of Geography, Northumbria University, Newcastle-Upon-Tyne, United Kingdom
[2] Romanian Academy, Institute of Speleology, Clinicilor 5, Cluj-Napoca, Romania
[3] Hungarian Academy of Science - Institute for Nuclear Research, Hertelendi Laboratory of Environmental Studies, H-4026 Debrecen, Bem ter 18/C, Hungary
[4] Institute of Geography Education, University of Cologne, 50931 Köln, Germany
[5] Institute of Geology and Mineralogy, University of Cologne, 50674 Köln, Germany

Correspondence to: Jack Longman (jack.longman@northumbria.ac.uk) and Daniel Veres (daniel.veres@ubbcluj.ro)

**Abstract.** Reconstructions of dust flux have been used to produce valuable global records of changes in atmospheric circulation and aridity. These studies have highlighted the importance of atmospheric dust in marine and terrestrial biogeochemistry and nutrient cycling. By investigating a 10,800-year long paleoclimate archive from the Eastern Carpathians (Romania) we present the first peat record of changing dust deposition over the Holocene for the Carpathian-Balkan region. Using qualitative (XRF core scanning) and quantitative (ICP-OES) measurements of lithogenic (Fe, K, Si, Ti) elements, we identify 11 periods of major dust deposition between: 9500-9100, 8400-8100, 7720-7250, 6350-6000, 5450-5050, 4130-3770, 3450-2850, 2100-1450, 800-620, and 60 cal yr BP to present. In addition, we used testate amoeba assemblages preserved within the peat to infer local palaeohydroclimate conditions. Our record highlights several discrepancies between eastern and western European dust depositional records, and the impact of highly complex hydrological regimes in the Carpathian region. After 6100 cal yr BP, we find that the geochemical indicators of dust flux become uncoupled from the local hydrology. This coincides with the appearance of millennial-scale cycles in the dust input and changes in geochemical composition of dust. We suggest this is indicative of a shift in dust provenance from local/regional (likely loess-related) to distal (Saharan) sources which coincide with the end of the African Humid Period and the onset of Saharan desertification.

## 1 Introduction

Atmospheric dust plays a major role in oceanic and lacustrine biogeochemistry and productivity (Jickells, 2005) by providing macronutrients to these systems (Mahowald et al., 2010). Furthermore, climatically dust plays a role in forcing precipitation (Ramanathan, 2001; Yoshioka et al., 2007) and in moderating incoming solar radiation. As such, reconstructions of past dust flux are an important tool to understand Holocene climate variability, biogeochemical cycles, and the planet's feedback to future changes in atmospheric dust loading.

The link between atmospheric circulation patterns and dust input has been studied intensively (Allan et al., 2013; Kylander et al., 2013; Marx et al., 2009; Le Roux et al., 2012) with clear evidence of climate variations linked with the dust cycle (Goudie and Middleton, 2006). Generally, dust is produced in arid zones (Grousset





and Biscaye, 2005) and may be transported thousands of miles before deposition (Grousset et al., 2003). In
addition, dust input into the atmosphere can increase significantly during droughts (e.g. Miao et al., 2007;
Notaro et al., 2015; Sharifi et al., 2015). As such, fluctuations in dust loading may be indicative of both regional
drying and long-distance transport (Le Roux et al., 2012).
Hydroclimatic fluctuations had a significant effect on the development of civilisations throughout the Holocene
(Brooks, 2006; deMenocal, 2001; Sharifi et al., 2015), especially on those which relied heavily on agriculture
and pastoralism, as was the case in the Carpathian-Balkan region (Schumacher et al., 2016). To understand the
impact hydroclimatic changes had on the population of an area of such importance to European history, high-
resolution palaeoclimate and palaeohydrological records are needed. This is especially important in the
Carpathian region, given the extensive loess cover in the area (Marković et al., 2015) - a fundamental factor in
sustaining high agricultural production. Additionally, the sensitivity of loess to moisture availability and water
stress during dry periods may turn this region and other surrounding loess belts into major dust sources (Kok et
al., 2014; Rousseau et al., 2014; Sweeney and Mason, 2013). This is particularly true under semi-arid (Edri et
al., 2016), or agriculturally-altered conditions (Korcz et al., 2009), as is the case with the major dust fields of
Eastern Eurasia (Buggle et al., 2009; Smalley et al., 2011; Újvári et al., 2012). Thus, the dust influx into the
Carpathian-Balkan region should be extremely sensitive to relatively small changes in precipitation rates. This
hydroclimatic sensitivity is enhanced due to the fact that the Carpathians and the surrounding lowlands are
located at a confluence of three major atmospheric systems; the North Atlantic, the Mediterranean and the
Siberian High (Obreht et al., 2016). Indeed, research appears to indicate the climate in Romania is controlled, at
least in part, by North Atlantic Oscillation (NAO) fluctuations (Bojariu and Giorgi, 2005; Bojariu and Paliu,
2001) but it is yet unclear how this relationship evolved in the past.
Multi-proxy and high resolution studies of palaeoenvironment in the region are still scarce, with most focusing
on reconstructing past vegetation changes (e.g., Feurdean et al., 2012). More recently, testate amoeba
(Schnitchen et al., 2006; Feurdean et al., 2015), pollen and diatoms (Magyari et al., 2009, 2013, Buczko et al.,
2013), and macrofossils (Gałka et al., 2016) have been utilised to elucidate the history of hydroclimatic
variability in the region. What is evident from these studies is the high inter-site variability, with clear
disagreements on timing and extent of wet and dry periods within a relatively small spatial distribution (e.g.,
two spatially close sites displaying differing precipitation trends as reported in Feurdean et al. (2008)). It is
possible that this variability reflects only site-related (including chronological) uncertainties, or is an indicator of
the impact of location at the contact of several climatic zones (Obreht et al., 2016). To determine this, the impact
of different modes of atmospheric (and moisture) circulation patterns and their imprint within paleoclimate
archives must be investigated through better regional coverage following high-resolution multi-proxy
approaches (e.g. Longman et al., in review.).
Our research provides a record of periodic dry and/or dusty periods in Eastern Europe as indicated by
reconstructed dust input, using an ombrotrophic bog from the Romanian Carpathians (Fig.1). As the only source
of clastic material deposited within ombrotrophic bogs is via atmospheric loading, such records have been used
convincingly as archives of dust deposition over the Holocene in Western Europe and Australia (Allan et al.,
2013; Kylander et al., 2013; Marx et al., 2009, 2010; Le Roux et al., 2012). To produce records of dust and/or
hydroclimate variability, both inorganic (Allan et al., 2013; Ross-Barraclough and Shotyk, 2003; Shotyk, 2002)



and organic (Booth et al., 2005; Lamentowicz et al., 2008; Morris et al., 2015; Swindles et al., 2010) proxies may be utilised (see Chambers et al. 2012 for a review).

Here we present the first record of dust input over the Carpathian Mountains, documenting changes in dust flux, source and intensity of deposition using the downcore lithogenic element concentrations from the Mohos ombrotrophic bog profile. The record covers 10,800 years of deposition over 9.5 m of peat, providing a much-needed high-resolution record for this region. Our research utilises both organic and inorganic proxies, with a high-resolution geochemical record of lithogenic elements (Ti, Si, and K), presented alongside the bog surface wetness as reconstructed using testate amoeba to understand dust source changes and the link between regional and extra regional hydroclimate variability and dust.

## 2 Materials and Methods

### 2.1 Geographical Setting

The Mohos peat bog (25°55' E; 46°05' N; 1050 m altitude, Figure 1) is located in the Eastern Carpathians, Romania, in the Ciomadul volcanic massif. The *Sphagnum*-dominated bog covers some 80 hectares, and occupies an infilled volcanic crater. There is no riverine inflow, which means that inorganic material deposited within the bog is predominantly derived via direct atmospheric transport. The climate is temperate continental, with average annual temperatures of 15°C and precipitation of 800 mm (Kristó, 1995). Surrounding vegetation is typical of this altitude in the Carpathians (Cristea, 1993), the bog being located at the upper limit of the beech forest, with spruce also found on surrounding slopes. Vegetation on the bog itself is diverse, with common occurrences of *Pinus sylvestris, Alnus glutinosa,* and *Betula pubescens*, alongside various *Salix* species (Pop, 1960; Tanţău et al., 2003).

The Mohos crater is related to volcanic activity from the Ciomadul volcano, which last erupted roughly 29.6 cal kyr BP in the neighbouring younger crater currently occupied by the Lake St Ana (Harangi et al., 2010; Karátson et al., 2016; Magyari et al., 2014; Wulf et al., 2016). The surrounding geology is dominated by andesites and dacites, occasionally capped by pyroclastic deposits and a thick soil cover.

### 2.2 Coring

A Russian peat corer was used to recover a 950-cm long peat sequence from the middle part of Mohos bog. The material consists mainly of *Sphagnum* peat and lacustrine sediments in the lowermost part. Upon recovery, the material was transported to the laboratory, described, imaged, and subjected to further analyses.

### 2.3 Sedimentological Parameters

Loss on ignition (LOI) was performed on ~1g (exactly 1cm$^3$) of wet peat, sampled at 2cm resolution. The peat was dried overnight at 105°C prior to ignition at 550°C for four hours. Weight loss after this combustion was used to calculate combusted organic material, followed by further combustion at 950°C for two hours to calculate total carbon content following carbonate removal (Heiri et al., 2001). The dry bulk density was determined from the known volume and the dry weight prior combustion.

### 2.4 Micro-XRF and MSCL Core Scanning

Non-destructive X-Ray fluorescence analysis was performed using an ITRAX core scanner (Croudace et al., 2006) at the University of Cologne (Institute of Mineralogy & Geology). The analytical resolution employed a 2-mm step size and 20 s counting time using a Cr X-ray tube set to 30 kV and 30 mA (and including a Si-drift



chamber detector). The method allows for a wide range of elements to be analysed, from which we have
selected Ti, K, and Si for further interpretation. To allow for clearer viewing, all XRF data sets were smoothed
using a 10-point adjacent averaging. Due to the methodological nature of XRF core scanning, the data are
presented as counts per second (cps) and are therefore considered as semi-quantitative. Due to poor counting
statistics, any values below 50cps should be treated with caution. Alongside geochemical analyses, magnetic
susceptibility (MS) was determined in 0.5 cm steps using a multi scanner core logger (MSCL, Geotek Ltd.) at
the University of Cologne equipped with a Bartington MS2E spot-reading sensor.
**2.5 ICP-OES**
To perform quantitative analysis of elements to allow inference of past dust flux, as well as to validate the
ITRAX data, ICP-OES analysis was carried out on 105 samples, at roughly 100-year resolution. These samples
were dried at 105°C overnight before homogenising using a pestle and mortar and then subjected to a mixed
acid ($HNO_3$: HCl: HF) total digestion (adapted from Krachler et al., 2002), for 40 minutes in a MARS
accelerated reaction system. The solution was then analysed using a Perkin Elmer Optima 8000 ICP-OES
system at Northumbria University. To monitor a potential instrumental drift, internal standard (1ppm Sc) was
added to all samples, and analysed alongside Ti. In addition, two Certified Reference Materials (CRMs) were
digested and analysed throughout the runs (Montana soil 2711 and NIMT/UOE/FM/001). Recoveries for both
CRMs were good for Ti, with average values of 85% and 79% respectively. Blanks with negligible Ti
contamination were run alongside the samples and CRMs.
**2.6 Calculating Dust Flux**
The dust flux delivered to an ombrotrophic bog via atmospheric loading may be calculated using the
concentration of a lithogenic element, such as Ti (Allan et al., 2013). Using the averaged occurrence of Ti in the
upper continental crust (UCC values from Wedepohl, 1995), the density of the peat as well as the peat
accumulation rate (PAR), the following formula may be used:

$$Dust\ Flux\ (\mathrm{g\ m^{-2}yr^{-1}}) = \left(\frac{[\mathrm{Ti}]_{sample}}{[Ti]_{UCC}}\right) \times density \times PAR \times 10000 (Eq.\ 1)$$

**2.7 Palaeoecological indicators**
A total of 44 samples of roughly 1cm$^3$ each were sampled along the peat profile for testate amoeba analysis. The
bulk samples were disaggregated and sieved according to Booth et al. (2010), prior to mounting in water on
slides. Two tablets of Lycopodium spores of known value were added to allow for calculation of test density.
For each sample at least 150 tests were counted, with identification of taxa following Charman et al. (2000). For
interpretation, two methods of determining wet and dry local depositional environments based on changes in
testate amoeba assemblages were used. Firstly, a transfer function (Schnitchen et al., 2006) already applied to
Carpathian bogs was used to reconstruct past variations in the depth of the water table (DWT). Secondly, the
main taxa were grouped into their affinity to wet or dry conditions according to Charman et al. (2000) and
plotted as a function of percentage.
**2.8 Chronology**
The age model for the Mohos peat record is based on 16 radiocarbon dates on bulk peat (collected over less than
1 cm depth interval per sample) consisting only of *Sphagnum* moss remains (Table 1). These analyses were
performed via EnvironMICADAS accelerator mass spectrometry (AMS) at the Hertelendi Laboratory of



Environmental Studies (HEKAL), Debrecen, Hungary using the methodology outlined in Molnár et al. (2013).
The $^{14}$C ages were converted into calendar years using the IntCal13 calibration curve (Reimer et al., 2013) and
an age-depth model (see Fig. 2) was generated using Bacon (Blaauw and Christen, 2011).

**2.9 Spectral Analysis**

Continuous Morlet wavelet transform was used to identify non-stationary cyclicities in the data (Grinsted et al.,
2004; Torrence and Compo, 1998). For this analysis, the lithogenic elemental data from ITRAX measurements
(Ti, K, Si and Fe) were interpolated to equal time steps of four years using a Gaussian window of 12 years.

**3 Results**

**3.1 Age Model and Lithology**

The Mohos peat profile is 950 cm long, and reaches the transition to the underlying basal limnic clay (Tanţău et
al., 2003). Between 950-890 cm the record is composed of organic detritus (gyttja) deposited prior to the
transition from a wetland into a bog. From 890 cm upwards, the core is primarily *Sphagnum*-dominated peat.
The age-depth model indicates the Mohos peat record covers almost 10,800 years of deposition, with the
uppermost layer (growing moss) of the peat dating to 2014. Thus, the resolution for ITRAX data average
~5yr/sample and for ICP-OES is roughly 100 yr/sample, respectively. The testate amoeba resolution is roughly
200 yr/sample. In the following, all quoted ages are in calibrated years before present (cal yr BP).

**3.2 Dust indicators**

**3.2.1 Ti, K, and Si**

Similar trends for the lithogenic elements Ti, and Si, and the mobile element K, are visible in the record (Fig. 3),
with 11 main zones of higher counts above typical background values present in all elements. These intervals
are further discussed as reflecting major dust deposition events, and are referenced in the remainder of the text
using the denotation D0-D10 (Fig. 3). We identify such events via the presence of a significant increase in one
or more of these elements. The lithogenic, and therefore soil and rock derived Ti and Si have previously been
used as proxies for dust input (e.g. Allan et al., 2013; Sharifi et al., 2015), whilst K covaries with Si ($r^2$=0.9945)
and so controlling factors on their deposition must be nearly the same. For these elements, the periods with
inferred non-dust deposition are characterised by values approaching the detection limit (150, 15 and 40 cps,
respectively). A short period of very high values for all elements (10,000, 1300 and 8000 cps, respectively) is
observed between 10,800-10,500 cal yr BP (not shown on diagram), reflecting the deposition of clastic
sediments within the transition from lake to bog at the onset of the Holocene. Zones of elevated values (D0-D5),
with average cps values roughly Ti=300, Si=30 and K=100 and persisting for several centuries each, occur
sporadically throughout the next 6000 years of the record, between 9500-9100, 8400-8100, 7720-7250, 6350-
6000 and 5400-5050 cal yr BP (Fig. 3). Similarly long periods, but with much higher element counts (Ti=800,
Si=60 and K=200 cps) occur between 4130-3770, 3450-2850 and 2100-1450 cal yr BP (D6-8). Two final, short
(roughly 100-year duration) but relatively large peaks (D9-10) may be seen in the last 1000 years, between 800-
620 cal yr BP (with values Ti=300, Si=40 and K=100 cps) and 75 cal yr BP to present (Ti=300, Si=80 and
K=400 cps, respectively).



### 3.2.2 Dust Flux

Using the quantitative ICP-OES values of Ti (in ppm), and equation 1, the dust flux can be calculated (Fig. 3). The ICP-OES Ti record shows very good correlation with the Ti data derived through ITRAX analysis, indicating the reliability of the XRF core scanning method even for such highly organic sediments (as already suggested by Poto et al., 2014), and validating its usage as proxy for deriving dust flux (Fig. 3).

The Ti-derived dust flux for most the record is below 1 g m$^{-2}$yr$^{-1}$, but with seven periods of dust deposition clearly identifiable for the last 6350 years, and several smaller fluctuations prior to that (mainly visible in the elemental data). The main peaks are similar in their timing to the ITRAX Ti trend, with three large peaks (dust flux >1.5 g m$^{-2}$yr$^{-1}$) located between 5400-5050, 2100-1450 and 800-620 cal yr BP, respectively (Fig. 3). Smaller peaks are present (dust flux 0-5-1 g m$^{-2}$yr$^{-1}$) at 6350-6000, 4150-3770, and 3500-2850 cal yr BP, respectively.

### 3.3 Density and Loss-on-Ignition (LOI)

Density values are relatively stable throughout the core, with all samples ranging between 0.06-0.1 g/cm$^3$. This trend is different from the organic matter values, which typically oscillate around 90-100% over the entire record. The very base of the record is however an exception, denoting the gradual transition from limnic clays to the peat reaching, with organic matter 80-90% between 10,800-10,000 cal yr BP. Very occasional intervals with lower organic matter content (roughly 85%) may be observed at 5400, 4100-3900, 3300-3200, 1900-1800 and 900-800 cal yr BP, respectively (Fig. 4).

### 3.4 Magnetic Susceptibility

The trend in magnetic susceptibility indicates that the peat is diamagnetic with values ranging from 0 to -2.5 (10$^{-5}$ SI). Slight increases up to -1.5 (10$^{-5}$ SI) may be observed 10,500-10,000, 7500-6500, and 2700-2450 cal yr BP. The highest values are in the topmost section of the record, with a sharp increase after 50 cal yr BP (Fig. 4).

### 3.5 Testate Amoeba

Two methods of clarifying the paleoclimate signal derived through investigating testate amoeba assemblages have been used (Charman et al., 2000; Schnitchen et al., 2006), with both indicating similar hydroclimatic trends. Reconstructions of depth to water table (DWT) values indicate three main trends within the record. The first, encompasses the time period between 10,800-7000 cal yr BP, and is characterised by highly fluctuating values, with four very dry periods (DWT ~20 cm) at 10,800-10,200, 9000-8800, 8600-7600, and 7400-6600 cal yr BP interspersed by wetter (DWT 15 cm) conditions (Fig. 4). After 7000 cal yr BP, values are much more stable, with DWT of 15cm until the final zone, the last 100 years, where DWT rises to 20cm. These fluctuations are in line with those seen in the wet/dry indicator species.

### 3.6 Wavelet Analysis

The wavelet analysis of K, Si and Ti show significant (above the 95% confidence threshold) periodicities between 1000-2000 years within the past 6000 years (Fig. 8). Prior to this, there appears to be no major cyclicity in the ITRAX data. Within periods which display raised ITRAX counts, shorter frequency (50-200 year) cycles are seen. These persist only for the period in which each element was enriched, particularly within the last 6000 years.





## 4 Discussion

### 4.1 Peat ombrotrophy

The relative intensities of the lithogenic elements analysed via ITRAX covary throughout the record (Fig. 3), despite their varying post-depositional mobility (Francus et al., 2009; Kylander et al., 2011). For example, the largely immobile Ti shows a very high correlation with that of redox sensitive Fe ($R^2 = 0.962$) and mobile K ($R^2 = 0.970$). This indicates the downcore distribution of these elements is mostly unaffected by post-depositional mobilisation via groundwater leaching and/or organic activity as documented in other studies (e.g. Novak et al., 2011; Rothwell et al., 2010). This, alongside the low clastic content (average organic matter of 91%), low density and domination of *Sphagnum* organic detritus, indicates the ombrotrophic nature of Mohos bog throughout time and validate the use of this record to reconstruct dust fluxes for the last ca. 10,000 years (Fig. 3).

### 4.2 The Dust Record

The record of inferred lithogenic (dust) input as indicated by Ti, K and Si documents 11 well-constrained periods of major and abrupt dust deposition (denoted D0-D10), with further small, short-term fluctuations (Fig. 3). The dust influx onto the Mohos peat was accompanied by decreases in organic matter (OM) as indicated from the LOI profile, and higher density values (Fig. 4), particularly over the intervals covered by events D5-D10. The major dust deposition events last from a few decades to hundreds of years (Fig. 3).

Firstly, it is noteworthy that five of the identified dust depositional events correlate to periods of Rapid Climate Change (RCC) as outlined by Mayewski et al. (2004) from the Greenland GISP2 record (Fig. 5). However, despite apparent hemispheric-scale influences, the dust events identified within Mohos record have little correlation to reconstructed European paleoclimate changes during Holocene. For example, the dust event between 3000-2700 cal yr BP falls within a cold period (Wanner et al., 2011). In contrast, the event between 860-650 cal yr BP is within the Medieval Climate Anomaly, a period of generally higher European temperatures (Mann et al., 2009) but also one of intense human impact on the environment thorough deforestation and agriculture (Arnaud et al., 2016; Kaplan et al., 2009). This indicates the dust depositional events are a result of a complex interplay of environmental conditions in the dust source areas, rather than simply reflecting warm or cold, or even wet or dry periods.

In addition to the North Atlantic, the impact of both the Mediterranean and the intertropical convergence zone (ITCZ) atmospheric systems influencing the Mohos dust record are apparent, including major climate changes in North Africa. D4 for example occurs within the chronological span of the 5900 cal yr BP event, a major cooling and drying period (Bond et al., 2001; Cremaschi and Zerboni, 2009; Shanahan et al., 2015). Increased dust influx is also recorded around 5300 cal BP (D5, Fig. 3) which roughly correlates with the end of the African Humid Period and onset of Saharan desertification (deMenocal et al., 2000). The lack of dust flux perturbations prior to 6100 yr BP, and their prevalence thereafter at Mohos appear to indicate a major shift in the controls of dust production and deposition at this time. The desertification of the Sahara around this time was the largest change in dust production in the northern hemisphere (see McGee et al., 2013; deMenocal et al., 2000).





Within our record, this initial dust flux increase was followed by a period of reduced dust loading, prior to a
rapid, and apparently major (highest dust flux values in the record prior to the most recent two millennia) event
at 5400-5000 cal yr BP. Regionally, Saharan dust in Atlantic marine cores strongly increased at this time, with a
140% rise roughly at 5500 cal yr BP (Adkins et al., 2006) and another rise by a factor of 5 by 4900 cal yr BP
(McGee et al., 2013). Furthermore, evidence from marine cores across the Mediterranean indicate decreasing
Nile output and increasing dust fluxes into the Eastern Mediterranean at this time (Box et al., 2011; Revel et al.,
2010). The correlation of these data to the Mohos record appears indicative of the region-wide impact of North
African desertification. It is noteworthy, as seen in Fig. 5, that the release of dust from the Sahara correlates well
with increasing frequency and intensity of dust fluxes at Mohos after 6000 cal yr BP, with all major (dust flux
>0.5 g m$^{-2}$yr$^{-1}$) Ti-derived dust flux peaks occurring after this time (Fig. 3). This period is the first indication of
the impact the Mediterranean climate and movement of the ITCZ has had on the Carpathian-Balkan region (as
simulated by Egerer et al. (2016) and Boos and Korty (2016)).  Indeed, intermittent intrusions of Saharan dust
over the Carpathian area have been well documented both through direct observations (Labzovskii et al., 2014;
Varga et al., 2013), and through provenance studies of past Saharan dust contribution within interglacial soils in
the region (Varga et al., 2016).
In addition to Saharan desertification, it is likely that early agriculture in the Carpathian-Balkan region has
contributed towards the increase in dust flux values at this time. It is known that advanced agriculture-based
societies inhabited the Carpathian area in the mid-Holocene (Carozza et al., 2012), with evidence of farming
seen in a number of pollen records (see Schumacher et al., 2016 for a compilation), including in Mohos itself at
the end of the Chalcolithic period (Tanţău et al., 2003). Since agriculture and soil erosion may be linked, it is
possible events D4 and D5 could also reflect to some extent dust input related to land disturbance by human
activities, on a regional scale. However, such evidence for agriculture, particularly in the proximity of Mohos is
limited to a few *Plantago* and cereal pollen (Tanţău et al., 2003), whilst the majority of pollen studies in
Romania at this time indicate no significant agricultural indicators (e.g. Magyari et al., 2010; Schumacher et al.,
2016; Tanţău et al., 2014). As such, it seems unlikely agricultural activity is behind such a large change in the
dust deposition record from Mohos.

### 4.3 Geochemical evidence for a dust provenance shift at 6100-6000 cal yr BP?

To better understand the nature of the shift in dust flux after 6100-6000 cal yr BP, a simple approach to
disentangling the geochemical makeup of the reconstructed dust load is discussed below. Figure 6 displays the
clustering of the lithogenic elements Ti and K (and Si, due to the similarity in the Si and K records) during dust
events D0-D10. The data appear to show three main types of dust (and presumably sources), one with high
values for both Ti and K (Type 1), one with relatively high values for K (Type 2), and one with relatively high
Ti compared to K (Type 3). The values for Ti-K correlation, average Ti and average K (in cps) are listed in
Table 2. Generally, the periods of no enrichment, and low K and Ti, do not show any correlation, indicative of
natural background and instrumental detection limits.
Type 1 deposition occurs only in D10, and is characterised by Ti-K gradient of nearly 1, indicating similar
values for both elements throughout the period, and a dust rich in both K and Ti. Type 2 deposition occurs in
several the dust events, particularly in D1-2, D4-5, and D7 (Fig. 8). The K enrichment which characterises these
events is, evidenced by the Ti-K gradients <1 and low (even negative in the case of D2) correlations between the



two elements.  Finally, Type 3 events (D3, D6, and D8-9) are characterised by an increased Ti-K gradient,
generally, around 0.2. The average Ti values during these events and the Ti-derived dust flux, are generally
highest in these periods (Table 2). These groupings would indicate similar dust sources within grouped events,
and may aid in identifying provenance.
Type 2 events typically occur in the older part of the record, except D7 (3400-3000 cal yr BP, Fig.8). Such
events are not visible in the Ti-derived dust flux values, indicative the reduced impact of Ti-bearing dust
particles deposited within the corresponding periods. The local rocks consist of K-rich dacites and pyroclastics
(Szakács et al., 2015), with relatively low Ti concentrations and enriched in K (Vinkler et al., 2007). Therefore,
the likely source of particulates deposited during these dust events is local or regional, with nearby (or even
distal) loess and loess-like deposits as another potential source, since loess sediments in south-eastern Europe
are generally not enriched in Ti (Buggle et al., 2008).
Type 3 events, conversely, appear Ti-enriched (Fig.8), with contribution from a source away from the low-Ti
dust of south-eastern European loess fields. These events typically occur after 6100 cal yr BP (Fig. 3). With the
periodic influence of the Mediterranean air masses in the region ((Apostol, 2008; Bojariu and Paliu, 2001)),
Saharan dust must be considered as a potential source area, since it appears to play a major role in dust input
into Europe today (e.g. Athanasopoulou et al., 2016). Geochemically, Saharan dust is typically Ti-enriched
(Nicolás et al., 2008). In particular, the Bodélé depression, the single-largest dust source in the Sahara, exhibits
extremely high Ti/Al and Ti enrichment (Bristow et al., 2010; Moreno et al., 2006). The Ti enrichment does not
show any regional trends, and so it is no use for determining exact source areas within the Sahara (Scheuvens et
al., 2013), but the presence of Ti-enriched dust appears to reflect a signal of Saharan influence. Consequently,
events of Type 3 may be considered to reflect, at least to a large extent, contribution of Saharan dust. Finally,
the single Type 1 event may be attributable to a mixing of sources, both local (resulting in high K) and distal
(resulting in high Ti), evidence for Saharan input and local soil erosion/deflation.
Previous work has indicated the input of Saharan dust in Eastern Europe, with evidence of such a source seen in
Carpathian loess (Újvári et al., 2012; Varga et al., 2013) and soil-forming dust (Varga et al., 2016).
Additionally, recent atmospheric satellite imagery has further confirmed the extent of Saharan dust outbreaks
and depositional events over central-eastern Europe (Varga et al., 2013). However, the lack of long-term dust
reconstructions in the region has so far precluded understanding of changing dust sources over the Holocene.
Previous studies across Europe indicate the complex input of dust from various sources over the mid-to-late
Holocene (e.g., Veron et al., 2014), but pertinently to our findings at Mohos, many examples exhibit a major
shift in dust sources at roughly 5000-7000 cal yr BP. In Belgium, Nd isotopes indicate a local source of dust
prior to input of European loess and Saharan dust after 6500 cal yr BP (Allan et al., 2013). This is echoed by
data from Le Roux et al. (2012) that indicate a major shift in the Nd isotopic composition at 6000 cal yr BP,
moving from a local to a mixed source, but with clear Saharan overprinting. The transition identified within the
Mohos dust record at 6100-6000 cal yr BP, therefore, appears to echo the appearance of a Saharan dust element
within other European bog-based dust reconstructions. However, it appears that input of Saharan dust was not
limited to the onset of North Africa desertification, as indicated by input of likely Saharan derived dust within





Mohos event D3 already by 7800-7200 cal yr BP. Further, even after 6100 cal yr BP, local sources still played a significant role, with D7 showing clear local or regional (e.g., loess-derived) signal.

D10 is interesting in that it appears to indicate even more K-rich dust sources. These values are similar in compositional gradient to the lake sediment deposited prior to the onset of peat formation in the early Holocene (Gradient of lake sediment samples (Average of samples pre-10,500 cal yr BP) = 0.7429, D10= 1.0637). Since the surrounding dacites and pyroclastics are K-rich (Vinkler et al., 2007), and the natural signal of erosion into the lake occurs prior to peat formation, it is reasonable to assume this period is indicative of local slope erosion. This is potentially due to the decline of the local forest and agricultural intensification, identified in the most recent sections of the Mohos pollen record (Tanţău et al., 2003). It is sensible to assume the local deforestation (visible around the Mohos bog as meadows for hay harvesting) has caused local soil erosion and increased dust production from very proximal sources (Mulitza et al., 2010). This is a clear sign of the persistent human impact at local to regional scale at the time (Giosan et al., 2012; Schumacher et al., 2016) that is also mirrored in the nearby Lake St Ana record (Magyari et al., 2009). As indicated by regional studies (e.g., Labzovskii et al., 2014; Varga et al., 2013; Vukmirović et al., 2004) high levels of Ti indicate Saharan input does not cease through this period, but that it is matched by high-K local sources. The apparent higher water table of the Mohos bog as implied by the TA record and the increased Ti contents indicative rather point towards an increasing Saharan influence rather than a major local dust source.

### 4.4 Correlation to other European dust records

Comparison to similar dust records from peat cores in Western Europe (Allan et al., 2013; Le Roux et al., 2012), and Atlantic margin sediments (McGee et al., 2013) reveals some interesting trends visible in all these records (Fig. 5), indicating comparable continent-wide controls on past dust flux. Specifically, the major dust event as seen at 5400-5000 cal yr BP in Mohos, and subsequent increase in number and intensity of dust events is comparable with an intensification of dust deposition over Europe after 6000 cal yr BP (Le Roux et al., 2012), with concurrent increases in dust flux in the Mid-Holocene documented in Belgium (Allan et al., 2013). The authors suggest a cool period (Wanner et al., 2011) as the cause of this dust increase. In addition to the reconstructed cool environments in Western Europe, this period is characterised by increased dust production in the Sahara (McGee et al., 2013), which is also likely to have played a role in the increasing dust flux over Europe. After 5000 cal yr BP, it appears Mohos and central-western European records show a more concurrent trend, with comparable dust peaks in the Swiss record (Le Roux et al., 2012) between 4100-3800, 3600-3050, 850-600 and 75 cal yr BP also present in Mohos, and a similar dust peak at 3200-2800 cal yr BP identified in another bog record from Bohemia (Veron et al., 2014).

Despite some similarities between the records, there is also significant variability, highlighting the difference between climatic controls in western and central Europe and those in south-eastern Europe. The disconnection between Mohos and other records is particularly clear for the early Holocene, with large dust flux peaks identified in Switzerland between 9000-8400 cal yr BP and 7500-7400 cal yr BP, when there is little evidence of dust input into Mohos. This discrepancy could be indicative of the east-west (Davis et al., 2003; Mauri et al., 2015; Roberts et al., 2012) and north-south (Magny et al., 2013) hydroclimatic gradients in Europe throughout the Holocene. As other studies indicate, south-eastern Europe was mostly disconnected (in terms of both precipitation and temperature) from the rest of Europe in the early-Mid Holocene (Davis et al., 2003; Drăguşin



et al., 2014), clearly indicated by the trend in the Mohos dust record. Since the Sahara had not undergone
significant desertification by this time, no clear correlation with western records may be made, hinting at more
local source for the earliest five dust events identified within the Mohos record (Fig. 3). In addition, the dust
events occurring during the early to mid-Holocene, which are not present in the Ti-derived dust record at
Mohos, are more likely related to local fluctuations in moisture availability, and Si and K rich soil dust.
**4.5 Palaeoecological Proxy Record**
To further investigate the difference between local and regional palaeoclimate signals within Mohos, and to
reconstruct the local hydroclimate conditions throughout the record, we use the fossil assemblages of testate
amoeba (TA). These data, alongside comparisons to existing Carpathian-Balkan and Mediterranean
hydroclimate reconstructions (Fig.7), may be used to further investigate the theory of a distal (most likely
Saharan) source for dust after 6100 cal yr BP. The earliest section in the TA record (10,800-6400 cal yr BP) is
characterised by fluctuating dry/wet periods, indicative of large shifts in the local hydroclimatic environment
(Fig. 4). The earliest identified dry period (10,800-10,000 cal yr BP) is linked to the shift away from a lacustrine
to a palustrine environment as a result of local drying. Three subsequent dry periods may be identified in the TA
record 9300-8800, 8500-8100, and 7800-7000 cal yr BP, all of which are also identifiable in the geochemical
dust record (D1-D3) via trends in K and Si. Between 10,200-7450 cal yr BP, dust flux at Mohos was low. Dust
events during this time are mainly present in the K and Si records (Fig. 3), or in OM and density parameters
(Fig. 4).
The first period of elevated dust proxies at roughly 10,300 cal yr BP (D0) correlates well with the 10,200 cal yr
BP oscillation (Rasmussen et al., 2007), previously linked to a drop in water levels at nearby Sf Ana Lake
(Korponai et al., 2011; Magyari et al., 2012, 2014). High *Difflugia pulex* and *Trigonopyxis arcula* values during
D1 as indicator taxa for dry conditions (Allan et al., 2013; Charman et al., 2000) appear to confirm local drying,
observed across much of the Mediterranean (Berger et al., 2016; Buczkó et al., 2013; Magyari et al., 2013, Fig.
7). The D2 and D3 events may also be observed in both the TA record and the geochemical dust record, with D2
attributable to the 8200 cal yr BP event (Bond et al., 2001), a paleoclimatic event already identified in other
local hydroclimate reconstructions (Buczkó et al., 2013; Magyari et al., 2013; Schnitchen et al., 2006). The
transition to the next wet period at 8000 cal yr BP also mirrors the dust record, with a deeper water table
occurring during the dust-free conditions between D2 and D3. This is prior to the bog undergoing dry conditions
between 7800-7000 cal yr BP, roughly in line with D3, drying which has previously been observed in Romania
(Gałka et al., 2016; Magyari et al., 2009, Fig 7). Due to the covariance between geochemical and
paleoecological proxies at this time, and the correlation to other local reconstructions, the early Holocene
section of the record indicates a close linkage of local hydroclimate and dust input. These dust events, therefore,
are likely the signal of remobilised material (Edri et al., 2016), from proximal or distal sources (including
perhaps from loess-derived sediments, at the foot of Ciomadul volcano) as the climate locally appears to become
more arid.
Between 6600-1200 cal yr BP, the TA indicate a shift to prolonged wet conditions, with only minor fluctuations
and no clear correlation to the geochemically-derived dust record, and so the dust events appear unrelated to
local drying within this time period (Fig. 7). This is indicative of a decoupling of the dust record from local



climate reconstructions, with dry phases common throughout the Mid-Late Holocene in other Romanian sites (e.g., Magyari et al., 2009; Schnitchen et al., 2006, Fig 7), and a distal dust source.

In the last millennium, there are two major dust events, with the first, D9, occurring between 850-650 cal yr BP. This episode falls within the late Medieval Warm Period, and could be related to human activity in the local area, as pollen from the Mohos bog indicates strong evidence for agriculture at roughly the same time (Tanţău et al., 2003). This may be seen in the intensity of the dust deposition at this time (dust flux >3 g m$^{-2}$yr$^{-1}$). D10, from 75 cal yr BP to present is certainly linked to such human influences, with the TA record echoing local studies, which display anthropogenically-altered conditions and intensive agriculture (Buczkó et al., 2013; Diaconu et al., 2016; Giosan et al., 2012; Magyari et al., 2009, 2013; Morellón et al., 2016; Schnitchen et al., 2006, Fig. 7). This appears to validate the geochemical approach used earlier, as intensive farming is likely to result in local dust mobilisation, with K-rich dust present at this time. This does not preclude Saharan input, however, as the dust is also Ti-rich.

**4.6 Periodicity**

To further understand the nature of the reconstructed dust events, cyclicity within the geochemical record was investigated using wavelet analysis (Fig. 8). The main elements of interest (Ti, Si and K) have no apparent cyclicity in the first half of the record (10,800-6000 cal yr BP), whilst the last 6000 years display clear centennial and millennial-scale cycles. A number of other studies have identified cyclicity shifts at this time (Fletcher et al., 2013; Jiménez-Espejo et al., 2014; Morley et al., 2014), related to North Atlantic variability, but so far mainly in western Mediterranean records. From 6000 cal yr BP onwards, the geochemical record at Mohos preserves two main cyclicities; one at ~1200-2000 years and the second at ~ 600-800 years (Fig. 8). A 715-775 year cycle has been determined as a harmonic of Bond event-related dry periods, present in other Northern hemisphere records (Springer et al., 2008) and in central Africa (Russell et al., 2003). The 1200-2000 years cycle, in contrast, is within the envelope of a 1750 year cycle observed within the western Mediterranean, in pollen (Fletcher et al., 2013) and Saharan dust (Debret et al., 2007; Jiménez-Espejo et al., 2014), which is attributed to changes in North Atlantic circulation.

Within the dust deposition events (Fig. 3), there is an overprinting of high-frequency cyclicity in the Ti record, especially within the last 5000 years (Fig. 8). These are particularly clear at 4200, 3400 and 1800 cal yr BP, but lower-confidence cyclicities may be seen in most dust deposition events. These are generally 100-200 years in length, and only last the extent of the dust outbreak. Cycles with lower than 140-year periodicities are possibly reflecting mainly background noise (Turner et al., 2016), but those longer in duration may be indicative of climatically forced fluctuations within drought events affecting the dust source areas. This suggests the reconstructed dust deposition events based on the Mohos record were not characterised by constant deposition of dust, but by periodic dust pulses. These short cycles could reflect solar forcing, with comparable 200-year cycles observed in humification profiles from peats (Swindles et al., 2012), sediments in the Baltic Sea (Yu, 2003), Pacific Ocean (Poore et al., 2004), and in North American peatland isotope records (Nichols and Huang, 2012). In many cases, such cycles have been linked to lower solar activity periods, low temperatures and increased precipitation oscillations, related to the De Vries/Suess 200 year cycle (Lüdecke et al., 2015). In the case of Mohos, these fluctuations have manifested themselves as shifts in dust deposition, and indicate the persistent effect solar dynamics has on all facets of climate system.





## 5 Conclusions

- The first record of Holocene drought and dust input in a bog from Eastern Europe documents eleven periods of high dust: 10,500-10,400, 9500-9100, 8400-8100, 7720-7250, 6150-5900, 5450-5050, 4130-3770, 3450-2850, 2100-1450, 800-620 and 60 cal yr BP to present.

- A major intensification in the number, and severity (as indicated by dust flux values) of dust events is observed after 6100 cal yr BP. Prior to this, dust flux was low, indicative of the local nature of the dust events recorded, and echoed by the testate amoeba assemblages, which displayed coeval drying of the bog surface and small (compare to the periods post 6100 cal yr BP) dust input. Thus, the dust record of the early to mid-Holocene may be more indicative of local-scale drought conditions.

- The timing of the major shift at 6100 cal yr BP is possibly related to the end of the African Humid Period, and the establishment of the Sahara Desert, pointing to significantly greater Saharan input within the regional dust loading after this time. This is corroborated by changes in cyclicity attributable to Saharan dust outbreaks, and a shift toward Ti-rich dust (a signal of Saharan rock and sediment) deposited onto the Mohos peat. Our data is the first such indication of the impact Saharan dust has had across Eastern Europe, in line with enhanced deposition of dust across the Mediterranean region. A tentative dust provenance analysis based on a simple geochemical approach to disentangle the composition of the dust has been applied to confirm this, with three main types of deposition documented, indicating the interplay between local/regional (mainly loess-derived) and Saharan dust sources over the Holocene.

- The most recent dust event, between 75 cal yr BP and today is geochemically indicative mainly of local erosion. This may be linked to the increasing human impact through deforestation, agriculture and recently tourism, and associated soil erosion, indicating a shift in the controls on drought and dust in the region.

**Author contribution.** J. Longman, D.Veres, V.Ersek and U.Salzmann designed the research, interpreted the results and wrote the paper. D.Veres, M.Bormann and F. Schabitz performed the fieldwork. J.Longman performed the ICP-OES, testate amoeba and statistical analysis. V.Ersek performed the wavelet analysis. M.Bormann, F.Schabitz and V.Wennrich perfomed the ITRAX and MSCL analysis. K.Hubay performed the [14]C dating. All authors approved the content of the paper.

**Competing interests.** The authors declare that they have no conflict of interest

**Acknowledgements**. We would like to thank Northumbria University for J.Longman's studentship. This is a contribution to the project PN–II-ID-PCE-2012-4-0530 "Millennial-scale geochemical records of anthropogenic impact and natural climate change in the Romanian Carpathians" and to the CRC806 "Our way to Europe" hosted at University of Cologne, Bonn & Aachen (subproject B2) granted by the DFG (German Research Foundation).

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

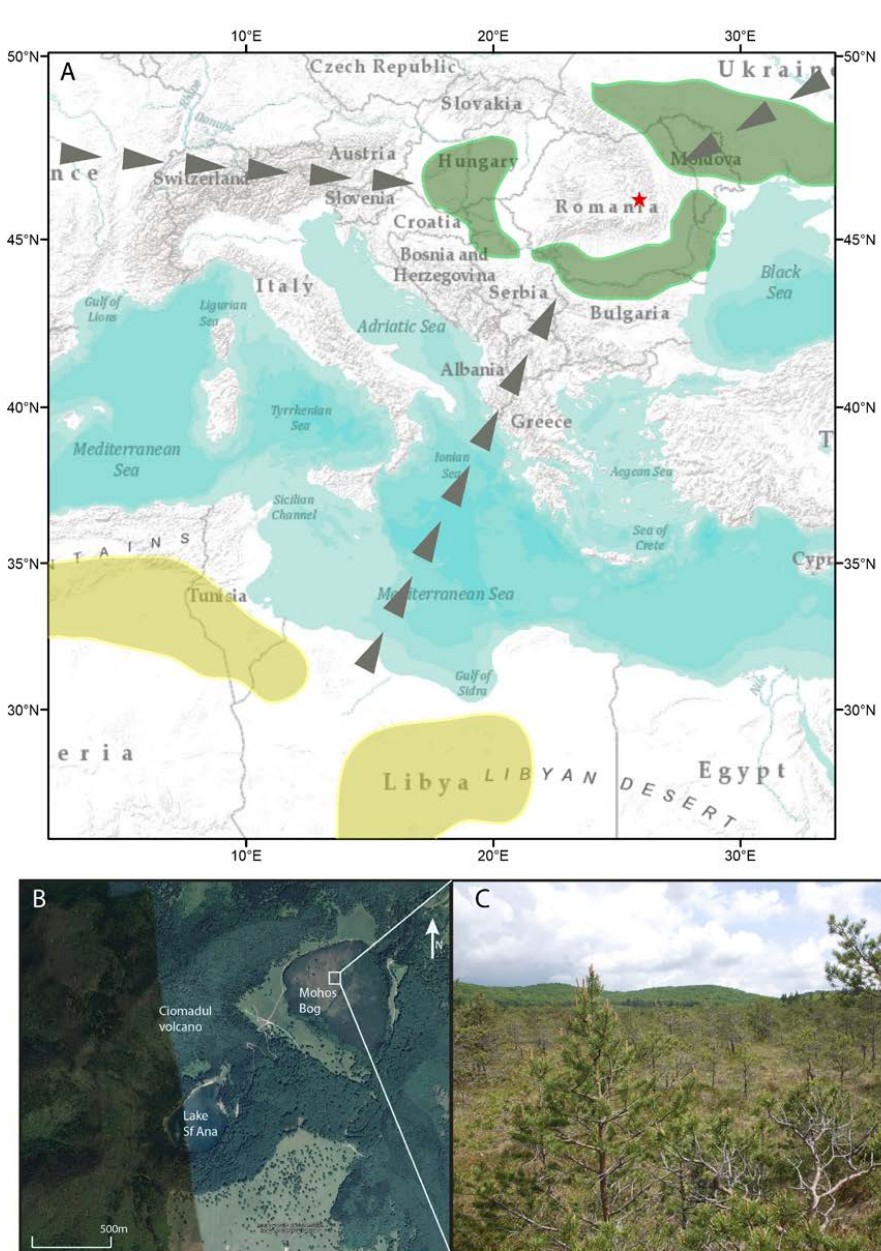

**Figure 1: 1.A: Figure 1: 1.A: Map of the Carpathian-Balkan region indicating location of Mohos peat bog (red star),**
**in the South-Eastern Carpathian Mountains. Predominant wind directions relating to air circulation patterns in the**
**area are indicated by black arrows. Major Saharan dust source areas are indicated in yellow (Scheuvens et al., 2013)**
**and local loess fields (including loess-derived alluvium) in green (Marković et al., 2015). 1.B: Map of Mohos and**
**neighbouring Lake Sf Ana, from Google Earth 6.1.7601.1 (June 10th 2016). Harghita County, Romania, 46°05' N ;**
**25°55' E, Eye alt 3.06 km, CNAS/Astrium, DigitalGlobe 2016. http://www.google.com/earth/index.html (Accessed**
**January 23rd 2017). Coring location within white box. 1.C: Photo of Mohos bog at the coring location with the crater**
**rim visible in the distance.**





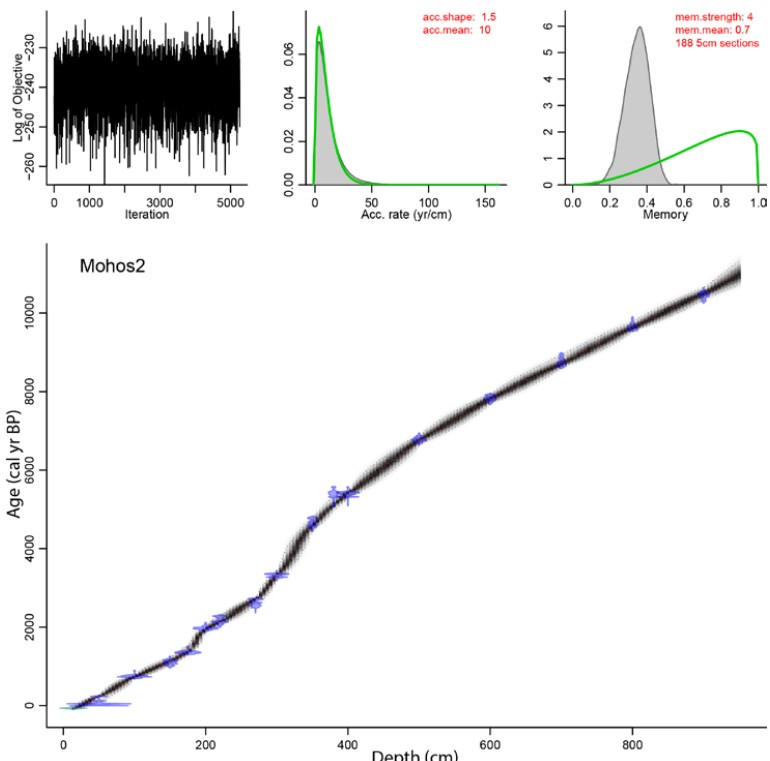

**Figure 2: Age-depth model of Mohos peat record, as determined via Bacon (Blaauw and Christen, 2011). Upper left graph indicates Markov Chain Monte Carlo iterations. Also on the upper panel are prior (green line) and posterior (grey histogram) distributions for the accumulation rate (middle) and memory (right). For the lower panel, calibrated radiocarbon ages are in blue. The age-depth model is outlined in grey, with darker grey indicating more likely calendar ages. Grey stippled lines show 95% confidence intervals, and the red curve indicates the single 'best' model used in this work.**





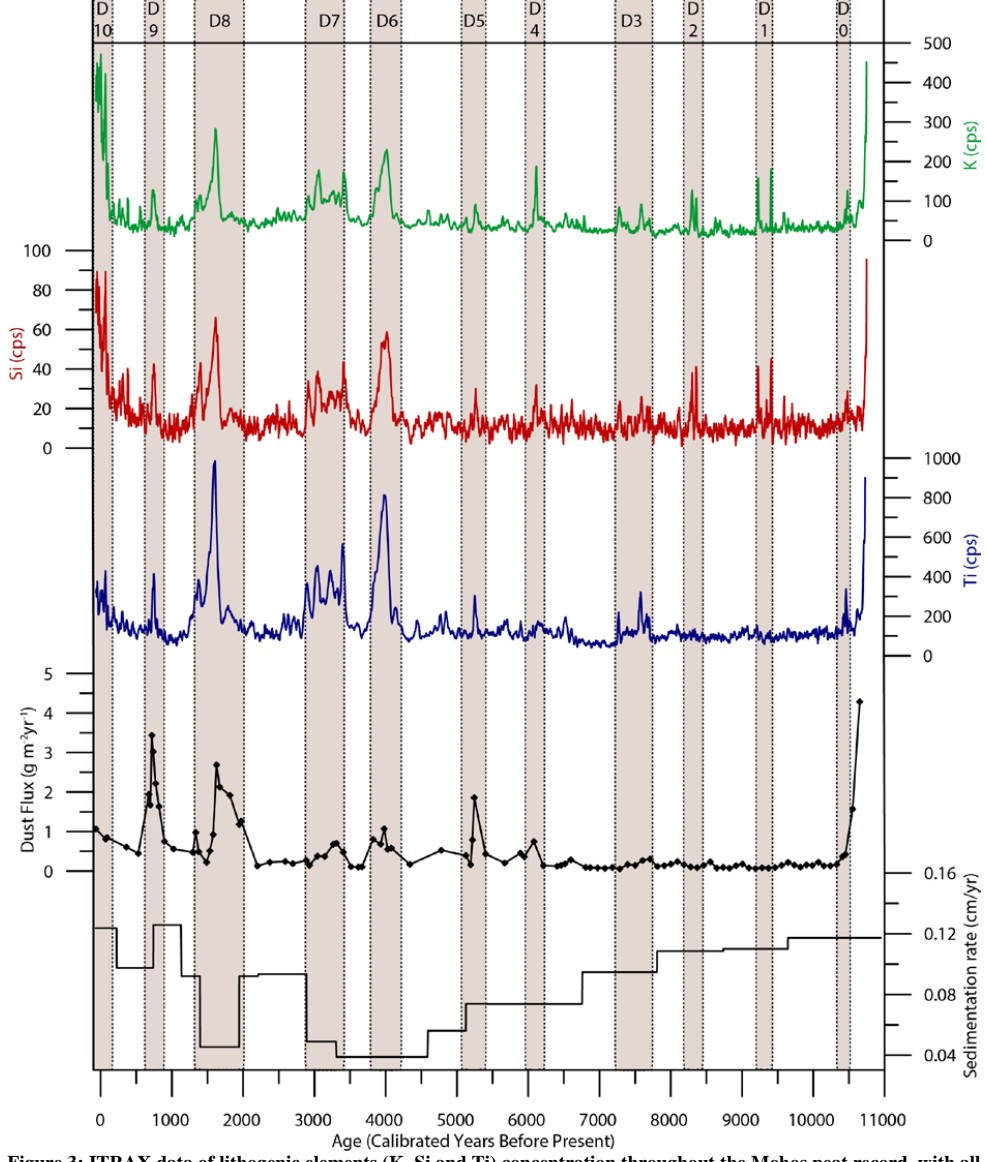

**Figure 3: ITRAX data of lithogenic elements (K, Si and Ti) concentration throughout the Mohos peat record, with all data smoothed using a 10-point moving average to eliminate noise. Alongside, dust flux as reconstructed by Ti concentration values, and sedimentation rate is presented. Dust events (D0-D10), as identified from increases in at least two of the lithogenic elements under discussion, are highlighted in brown.**





**Figure 4: Comparison of Ti-derived dust flux record with magnetic susceptibility (10$^{-5}$ SI), wet and dry TA indicator species % values, reconstructed Depth to Water Table (DWT), organic matter and dry density. Vertical bars as in Fig. 3.**





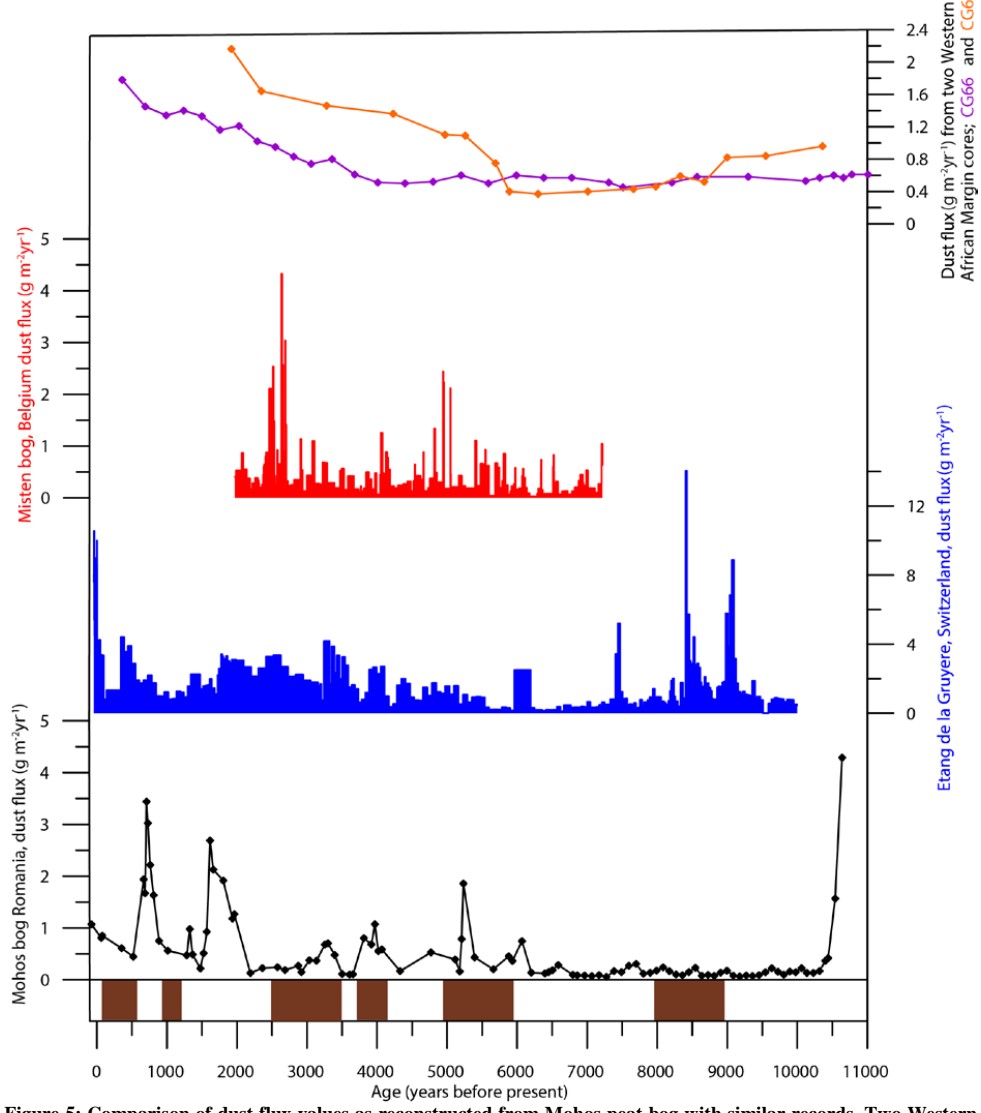

**Figure 5: Comparison of dust flux values as reconstructed from Mohos peat bog with similar records. Two Western African dust flux records (GC 68 and 66) from marine cores (McGee et al., 2013), are presented alongside bog-based records from Misten bog in Belgium (Allen et al., 2013) and Etang de la Gruyere in Switzerland (Le Roux et al., 2012) respectively. These are presented alongside the dust flux record from Mohos (lower panel). Also shown, in brown, are periods of Rapid Climate Change derived from Greenland Ice (Mayewski et al., 2004).**





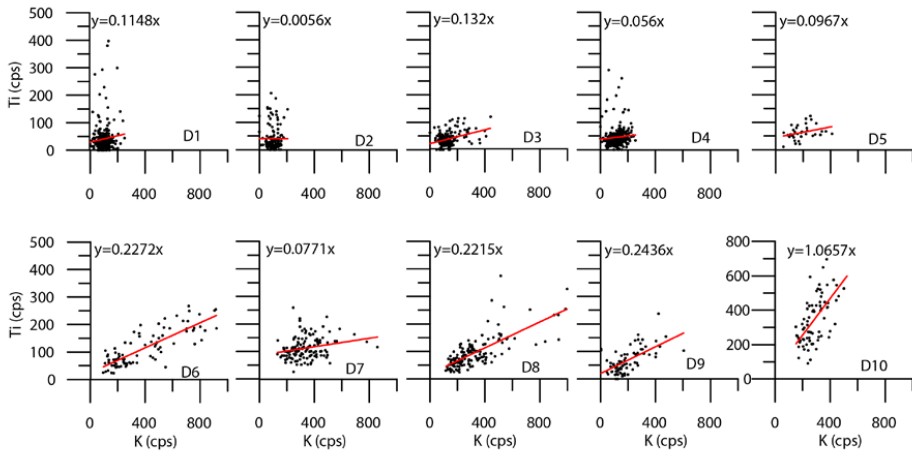

880

881 **Figure 6: Correlation graphs and gradients of Ti versus K for each of the dust events (D1-D10)**

882




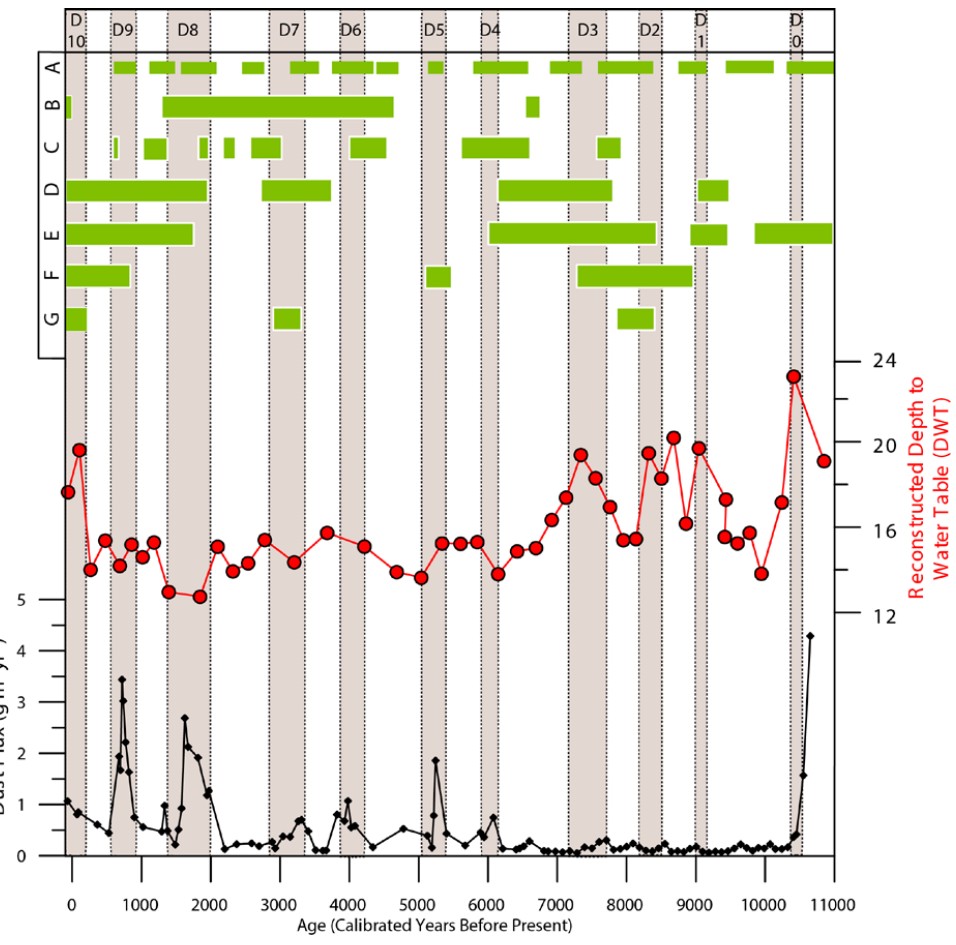

**Figure 7: Comparison of dust events and bog wetness as reconstructed from the Mohos record, to regional hydroclimate reconstructions. Data presented via green bars is drought/dry/low lake periods from the following publications. A: (Magny, 2004), B: (Cristea et al., 2013), C: (Galka et al., 2016), D: (Magyari et al., 2013), E: (Buczkó et al., 2013), F: (Magyari et al., 2009), G: (Schnitchen et al., 2006). These are presented alongside the Mohos Ti record, Ti-derived dust flux.**





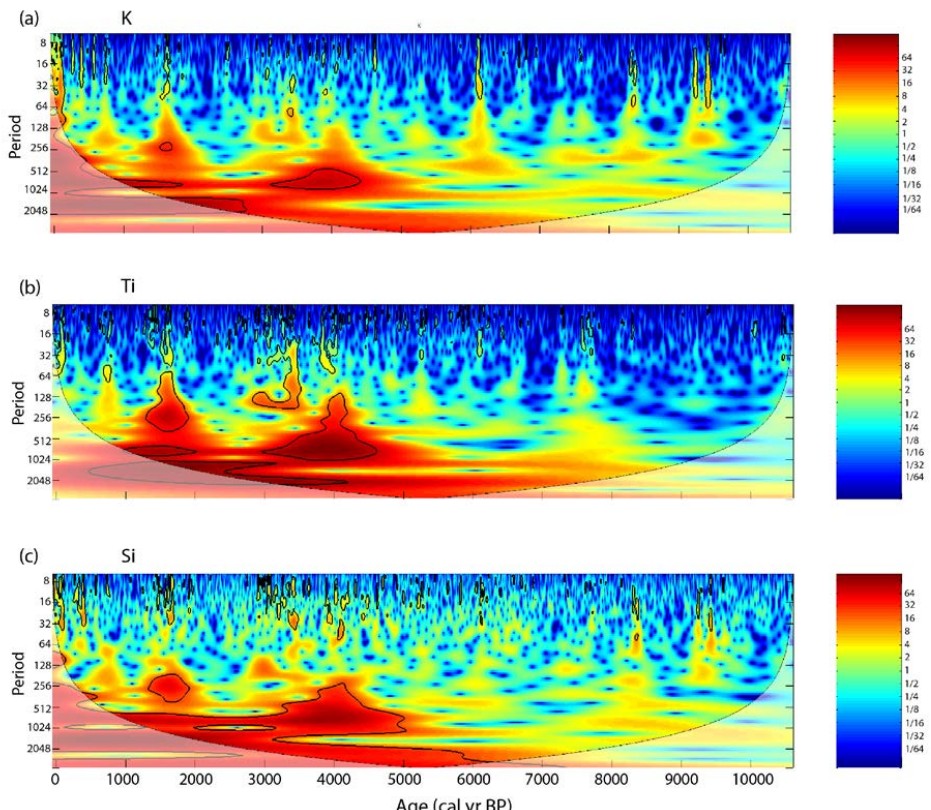

**Figure 8: Spectral analysis of Mohos ITRAX geochemical data for A: Ti, B: K, C: Si. Areas outlined in black are significant at the 95% confidence level. Shaded area indicates the cone of influence, outside of which results may be unreliable.**



**Table 1: Radiocarbon dates used to build the age model for Mohos peat record.**

| Lab ID | Depth | ¹⁴C age (yr BP ± 1σ) | Calibrated age (cal yr BP ± 2σ) | Dated material |
|---|---|---|---|---|
| DeA-8343 | 50 | **37±18** | 37-65 | bulk peat |
| DeA-8344 | 100 | **838±19** | 700-785 | bulk peat |
| DeA-10111 | 150 | **1174±28** | 1049-1179 | bulk peat |
| DeA-10112 | 175 | **1471±26** | 1309-1399 | bulk peat |
| DeA-8345 | 200 | **2022±21** | 1921-2007 | bulk peat |
| DeA-10137 | 225 | **2155±27** | 2048-2305 | bulk peat |
| DeA-10138 | 280 | **2530±28** | 2495-2744 | bulk peat |
| DeA-8346 | 300 | **3112±23** | 3249-3383 | bulk peat |
| DeA-10139 | 350 | **4110±31** | 4523-4713 | bulk peat |
| DeA-10140 | 380 | **4641±54** | 5282-5484 | bulk peat |
| DeA-8347 | 400 | **4638±26** | 5372-5463 | bulk peat |
| DeA-10141 | 500 | **5949±36** | 6677-6861 | bulk peat |
| DeA-10142 | 600 | **6989±43** | 7785-7867 | bulk peat |
| DeA-8348 | 700 | **7909±33** | 8600-8793 | bulk peat |
| DeA-10143 | 800 | **8687±45** | 9539-9778 | bulk peat |
| DeA-8349 | 900 | **9273±36** | 10369-10571 | bulk peat |




**Table 2: Ti-K correlation ($R^2$), alongside average cps for K and Ti for each of the dust events as identified within the**
**Mohos core.**

| Dust Event | D1 | D2 | D3 | D4 | D5 | D6 | D7 | D8 | D9 | D10 |
|---|---|---|---|---|---|---|---|---|---|---|
| **Ti-K Correlation ($R^2$)** | 0.09 | 0.00 | 0.37 | 0.08 | 0.33 | 0.82 | 0.23 | 0.77 | 0.62 | 0.63 |
| **Average Ti (cps)** | 106.51 | 106.41 | 121.51 | 106.19 | 193.06 | 431.80 | 308.44 | 268.04 | 197.07 | 288.68 |
| **Average K (cps)** | 48.68 | 57.80 | 52.33 | 32.05 | 63.14 | 124.3 | 108.2 | 76.04 | 67.71 | 349.8 |
