# Peer review of "Periodic input of dust over the Eastern Carpathians during the"

_Climate of the Past, 2017_

## Referee Comment (RC1) · Anonymous Referee #1 · 24 Feb 2017

The manuscript presents profiles of major elements and paleoecological indicators from an ombrotrophic peat bog in Romania. The paleoclimate records are associated to mineral dust and the discussion is focused around possible interpretations that try to disentangle local from distal signals. The authors interpret the history of the site in function of a superposition of changes on diverse spatial scales, from that of local hydrology to the large scale / hemispheric patterns derived from Greenland ice cores or North Atlantic marine sediment records. The topic is of interest to the paleoclimate community. The work appears well structured and its presentation in the manuscript is generally clear. I think that a better discussion of uncertainties would improve the manuscript.

[Figure]

General comment

My comment about improving the discussion of uncertainties is articulated in two parts, that have to do with (1) the quantification of the dust flux and (2) the attribution of potential sources.

(1) For the purpose of estimating the dust flux, only the concentrations of Ti from ICP-OES are used, although a semi-quantitative comparison is carried out against three major elements counts from XRF. Other studies trying to estimate dust from peat bog records discussed the uncertainties related to this issue (e.g. Marx et al., 2009; Kylander et al., 2016). Please discuss more in detail these aspects, and if possible provide some estimates of the uncertainties.

(2) The attribution of potential sources is based on a simple analysis of correlations between major elements counts, as discussed by the authors, and is also supported by the interpretation of the evidence from indicators such as testate amoeba and pollen. Nonetheless, it would only be by looking at dust size distributions at the same time, that one could derive more firm conclusions about distal versus local contributions to the dust budget (e.g. Mahowald et al., 2014). If possible, include this kind of data, otherwise please discuss this aspect in the text.

Specific comments

p. 3, 103-106. Please describe how the cores were packed and stored in the phase between recovery and analyses.

p. 4, 120-121. Later in the text (line 179) you also mention different detection limits for the different elements. Please provide all this kind of information in the same place in the text, and try to mark it visually in the plots.

Figures 3 & 4. I think a slight confusion can arise because of the way some of the records are organized between these two figures. For instance, it would be useful to see in Figure 3 the Ti concentration profile from ICP-OES along with the major elements counts from XRF, rather than directly the dust flux which weights in the sedimentation rate and bulk density profiles. On the other hand it would be nice to have all the factors defining the dust profile in the same figure, i.e. Ti concentration profile from ICP-OES, sedimentation rate and bulk density profiles.

p. 4, 126. What is the depth span of each sample? i.e. was the full core analyzes, or just portions of it?

p. 5, 160. Harmonize with what you say at line 118.

p. 5, 175. What do you mean by significant? Did you apply some statistical test? Otherwise perhaps change with "visible".

p. 6, 190. Please discuss the uncertainties in estimating the dust flux.

p. 6, 208-211. It is not clear at this point what is the contribution of this kind of analysis to the work.

p. 7, 238. The dust "Dn" events are selected based on the occurrences of at least one of the elements form the XRF scan, so it is a bit weird to go through the text until this point with some apparent inconsistency between the discussion of peaks D0 to D3, which are not evident and sometimes in anti-phase the what you call dust record. Please either change your definition of dust event (D) or harmonize the text.

p. 7, 244-245. As you discuss below, there is not a correlation, so please rephrase with something like "we compare the timing of the identified dust prepositional events with periods ..."

p. 7, 260. "Appear to indicate": maybe it would be better "are consistent with", in relation to my comment about the missing information on dust particle size distributions. Please include this kind of data if possible, otherwise add a discussion paragraph about this issue.

Figure 5. Check the units of the upper curves (the two cores from McGee et al., 2013):

[Figure]

I think you reported the values with the wrong units, which I think are g/cm2/kyr in the original publication, so there is a factor ten difference. In fact, on this scale you have the same dust flux, if not lower, in the North African plume and Belgium or Romania, which would be weird.

p. 8, 266-268. It is not very clear what is the relation between these two studies, please rephrase.

p. 8, 279-289. Again, particle size distributions would help clarify these issues.

p. 8, 291-298. Earlier in the text you mentioned the different mobility of these major elements, and how the similarity of their profiles supports the ombrotrophic nature of the peat bog. Please clarify how this is coherent with your analysis here, which is instead based on the difference between the same elements.

p. 9, 310-313. It would be interesting to compare Type 2 events with the background signature of non-D periods.

p. 9, 314. "Fig. 8" should probably be "Fig. 6", please check.

p. 12, 429. Was the data interpolated somehow before performing wavelet analysis? What is the pace / temporal resolution of the time series fed to the wavelet analysis software?

References

S.K. Marx, H.A. McGowan, B.S. Kamber, Long-range dust transport from eastern Australia: a proxy for Holocene aridity and ENSO-type climate variability, Earth Planet. Sci. Lett., 282 (2009), pp. 167–177.

Malin E. Kylander, Antonio Martínez-Cortizas, Richard Bindler, Sarah L. Greenwood, Carl-Magnus Mörth, Sebastien Rauch, Potentials and problems of building detailed dust records using peat archives: An example from Store Mosse (the "Great Bog"), Sweden, Geochimica et Cosmochimica Acta, Volume 190, 1 October 2016, Pages

156-174, ISSN 0016-7037, http://dx.doi.org/10.1016/j.gca.2016.06.028.

Mahowald, N., S. Albani, J. F. Kok, S. Engelstaedter, R. Scanza, D. S. Ward, and M. G. Flanner (2014). The size distribution of desert dust aerosols and its impact on the Earth system. Aeolian Research, 15, 53-71, doi: 10.1016/j.aeolia.2013.09.002.

---

## Editor Comment (EC1) · D.-D. Rousseau (Editor) · 25 Feb 2017

Dear Authors, Reviewer 1 posted his review raising some issues. As we are during the discussion phase I encourage you to take this opportunity to exchange with reviewer 1 by posting a reply. All the very best

denis-didier Rousseau

Climate of the Past Co-Editor in Chief

---

## Short Comment (SC1) · 27 Feb 2017

Firstly, we thank the reviewer for their constructive comments and suggestions and we will endeavour to address them all. With regards to the reviewer's general comments, discussion of uncertainty within peat-based dust records is something we will add to the manuscript, and we will attempt some grain size analyses of the events themselves. However, the low mineral content of the peat may make these analyses difficult. Regarding the more specific comments, we will adapt figures accordingly, ensure all text is clear and coherent, and fill any gaps regarding interpretation or explanation (e.g. type of interpolation for wavelet analysis).

---

## Referee Comment (RC2) · N. Fagel (Referee) · 1 Mar 2017

The ms presents a very interesting and complete dataset on Holocene dust reconstruction from East European peat record. The ms is well organized with clear aims. The text is too the point. The approach is quite innovative. See the attached pdf with several remarks and/or suggestions on the text and the figures. For my point of view, two mains points should be completed. First on the methodological approach I suggest to present the calibration between ITRAX and ICP-OES data to better constrain the proposed approach. I will also discuss the potential influence of porosity/humidity changes in the measured peat section on the number of ITRAX counts. Indeed elemental ITRAX data are often normalized, for instance by the total

number of measured counts in order to take into changes in peat density/porosity but also on surface irregularities. I would explain why this approach is not useful. Second I would complete the text about the relationships between the identified dust events, the wet/dry, cold/warm conditions and the related figures. Hoping the review will be helpful to improve this interesting manuscript. Sincerely, Nathalie Fagel

Please also note the supplement to this comment:
http://www.clim-past-discuss.net/cp-2017-6/cp-2017-6-RC2-supplement.pdf

---

## Editor Comment (EC2) · D.-D. Rousseau (Editor) · 3 Mar 2017

Dear authors,

As I suggested you with the first review, please take the opportunity of the discussion phase to post a general comment about this second review.

All the very best

denis-didier Rousseau

Climate of the Past Co-Editor in Chief

---

## Short Comment (SC2) · 6 Mar 2017

Dear Nathalie,

Thank you very much for your kind words and review, in particular the valuable suggestions regarding our manuscript. With regards to your main comments, you raise a good point by suggesting normalisation of the ITRAX data; indeed this approach has been used on similar datasets in the past. We will implement this suggestion as best we can (likely by using the inc and coh ITRAX values), along with a number of smaller comments regarding data presentation and interpretation.

Further, it is an interesting suggestion to investigate more fully the relationships be-

tween the identified dust events, and concurrent wet/dry and cold/warm periods. We will analyse any such relationships, and attempt to clarify exactly what the data is showing with regard to climatic trends.

Again, thank you for the review; it will certainly improve the manuscript.

Best regards,

Jack

---

## Author Comment (AC2) · 11 Apr 2017

N. Fagel (Referee) nathalie.fagel@ulg.ac.be The ms presents a very interesting and complete dataset on Holocene dust reconstruction from East European peat record. The ms is well organized with clear aims. The text is too the point. The approach is quite innovative. See the attached pdf with several remarks and/or suggestions on the text and the figures. For my point of view, two mains points should be completed. First on the methodological approach I suggest to present the calibration between ITRAX and ICP-OES data

to better constrain the proposed approach. I will also discuss the potential influence of porosity/humidity changes in the measured peat section on the number of ITRAX counts. Indeed elemental ITRAX data are often normalized, for instance by the number of measured counts in order to take into changes in peat density/porosity but also on surface irregularities. I would explain why this approach is not useful.

R1: We thank Dr. Fagel for the comprehensive review of our work. We fully agree, and have addressed both comments in the revised manuscript. Firstly, we now present the correlation, and significance of said correlation, between ICP-OES and ITRAX. Secondly, we have normalised our ITRAX data (by incoherent + coherent scattering) with respect to the scattering effect of density, porosity and organic matter changes. As such, all data is now presented as normalised counts per second, and all corresponding further analysis has been performed on such data. Please see the comments below for exact location of added text, and adapted figures.

Second I would complete the text about the relationships between the identified dust events, the wet/dry, cold/warm conditions and the related figures.

R2: We have also addressed these issues; see below for text additions, and figure adaptations

Hoping the review will be helpful to improve this interesting manuscript. Sincerely, Nathalie Fagel Please also note the supplement to this comment: http://www.clim-past-discuss.net/cp-2017-6/cp-2017-6-RC2-supplement.pdf Specific comments from attached Supplement: Line 26 unclear older or younger than 6,1 kyr?

R3: "Since" 6100 yr BP; manuscript has been adapted to reflect this

Lines 125-126 For my pint of view those data must be more exploited in the ms. Why do you not present the calibration between ITRAX and ICP data?

R4: This is a valid point and we have performed such measurements. Due to the differences in resolution of the data sets, we have performed a Gaussian interpolation

at 100 yr steps, with a window of 300 years in order to put both records on the same timescales. The correlation (Pearson's R, n=105) of this is 0.2649, with a p-value of 1.416×10-8, indicating a significant relationship between the two methods. The apparent weakness of the correlation is likely a function of the early half of the record, where low values are observed in both methods, and so the correlation is simply noise, and so we do not expect a perfect correlation. We have attached a graph to display this here: Such correlation data has also been added to the manuscript on lines 203-206: "To facilitate comparison, we bring both records on the same timescale using a Gaussian interpolation with 100 year time steps and a 300 year window. Pearson's r=0.2649, with a p-value of <0.001, indicative of a significant correlation (See SI 4)."

Lines 164-165 You should give the age of the lithological transition. I would suggest to add a schematic lithological column in regard with Fig 3.

R5: The age of the lithological transition has been added to the manuscript on line 173: "transition from a wetland into a bog, at approximately 10,330 yr BP" A lithological description, and full core image was included in the supplementary information: SI1 & SI2

Lines 175-176 (1) The identification of the D events must be precise. It is not clear why you merge some peaks into one D event (e.g., D1, D8).

R6: Dust events were identified based on the increase in two or more of the ITRAX-derived element concentration profiles (see caption on Fig. 3). For each element, a cutoff point was identified, with K=0.001, Si= 0.001 and Ti= 0.004. There are two exceptions, the very earliest section of the core, close to the minerotrophic-ombrotrophic transition, and in the most recent 1000 years, where the noise is higher. Some dust events do indeed appear to merge peaks, this is because we believe they are related to the same event, as they are very close to each other chronologically. An explanation of the denotation of dust events is now included on lines 183-187: "Such zones are identified as an increase in two or more of the elements above the background deposition

(K > 0.001, Si > 0.001 and Ti > 0.004, see dashed line on figure 3). These intervals are further discussed as reflecting major dust deposition events, and are referenced in the remainder of the text using the denotation D01-D10 (Fig. 3). Two exceptions, at the base of the core, close to the transition from lake to bog, and the last 1000 years, due to high noise, are not highlighted."

(2) What is the age uncertainty for each interval according to your age model? It is important to precise since some events cover less than 200 years.

R7: The age model uncertainties range from approximately 20 years at the uppermost sections of the core, to around 150 years by the base of the core (See Fig. 2 and Table 1). As such, events lasting 200 years should be discernible at this resolution. This has been added to the manuscript on lines 175-176: "Age model uncertainties range from 20 years in the uppermost sections to 150 years at the base of the core."

(3) On figure 3, I would suggest to add a secondary scale for dust flux in order to check for smaller peaks.

R8: We have added such a figure to the supplementary information- see SI 3.

Line 192 You must give more arguments to support this assumption. Do you calculate the correlation coefficient between the two datasets? I would add the figure, even as supplementary data. See also notes on fig. 3 and 4.

R9: Such data has been added to lines 203-206. See above for our description of the process, and a simple graph of said comparison (also added to the manuscript as SI 4).

Line 204 add a synthetic lithological column on Fig; 4 to indicate the transition

R10: See SI 2 for lithological column

Line 225 text unclear.

R11: Text has been altered on lines 246-247: "These persist only for the period in

which each element is enriched, with such cycles particularly evident within the last 6000 years."

Line 230 Text unclear. Indeed the similar profiles in the three elements emphasize that there is no post-depositional elemental mobility: all elements behave as conservative in the studied peat. I would reorganize the paragraph.

R12: We have inserted the following into line 254: " indicating the conservative behaviour of such elements in the studied peat."

Lines 247-248 For your record I would suggest to describe deeper the relationships between air temperature and/or humidity and dust flux. In the Misten dust record most dust peaks coincide with cold events.

R13: We agree such a discussion improves the manuscript, and so we have adapted lines 268-276 to reflect this: "For example D8, between 3450-2800 cal yr BP falls Europe-wide cold period (Wanner et al., 2011). Such cold-related dust deposition has been observed previously in western Europe. However, within Mohos such a compassion may not be drawn for the majority of dust events. For example, event D9 (860-650 cal yr BP) occurs during the Medieval Climate Anomaly, a period of generally higher European temperatures (Mann et al., 2009) but also one of intense human impact on the environment through deforestation and agriculture (Arnaud et al., 2016; Kaplan et al., 2009). Furthermore, such events within the Misten record (Allan et al., 2013) were also linked to low humidity, whereas the Mohos TA (Fig. 4) record indicates locally wet conditions. This suggests that dust depositional events in this region are a result of a complex interplay of environmental conditions in the dust source areas, rather than simply reflecting locally warm or cold, or even wet or dry periods."

Lines 260-261 also observed in other bog records (misten, EGR)

R14: We have revised the text on lines 284-285 to reflect this: "a major shift in the controls of dust production and deposition at this time, a shift observed in peat-derived

dust records from Western Europe (Allan et al., 2013; Le Roux et al., 2012)."

Line 299 it could reflect the minerotrophic stage of the peat, with no meaining on dust flux.

R15: Type 1 deposition only occurs within D10- the most recent dust event, not D0, the event close to the minerotrophic stage of peat. We agree however, that D0 may therefore not be a dust event, but related to minerotrophy, and the transition to ombrotrophy, and so we now do not discuss it as a dust event.

Line 344 not clear with double (.

R16: Text adapted line 372: "(Gradient of samples pre-10,500 yr BP = 0.7429, D10= 1.0637)."

Lines 353-356 D10 is older or very close to the minerotrophic/ombrotrophic transition. I would not give to much interpretation in term of dust flux for this first D event. The peat at that time may be affected by surface runoff.

R17: As mentioned above, we agree with the reviewer's interpretation regarding D0, and so we do not discuss it as a dust event.

Line 373 you must first remove signal from volcanic eruption observed in EGR record (see Le Roux et al., 2012).

R18: Text has been adapted to reflect this in lines 401-403: "with a large dust flux peaks identified in Switzerland between 9000-8400 cal yr BP with other volcanic eruption-related peaks (See Fig. 5), when there is little evidence of dust input into Mohos."

Lines 414 and 418 Fig .7 - But there are still several meter variations in water table: it is significant, no? Moreover most D events during this interval coincide with shallower water table. Such wetter conditions may limit local erosion and favor distal input. We observe the same relationship for the Misten bog (see Allan et al., 2013).

R19: The DWT values are in cm, something which has now been indicated on the

figures. As such, the variations in this period only indicate changes of roughly 2-5 cm change at most. The second point regarding such wet conditions favouring distal deposition is valid, however, and so we have adapted the text accordingly on lines 448-449: "Such wetter conditions also limit local drought-related erosion, and so may be further evidence of distal dust input at this time (Allan et al., 2013)."

Lines 427-428 you may add that the local input erase any distal input in that case.

R20: The text has been adapted lines 458-459: "local dust mobilisation, with K-rich dust present at this time, with local input potentially erasing some distal signals."

Line 436 The time interval is quite short to evidence cyclcity > 1 kyr. i would suggest to keep "millenial cyclicity".

R21: Text has been adapted in lines 467 and 470 to reflect this

Lines 460-463 I would suggest to rewrite point 2. Here it overlaps with the point 3. I would say that two intervals are identified wit different major dust control: 1) dust input over the older interval reflects more local conditions; 2) distal input for the younger interval .

R22: In order to improve the clarity , we have re-written this point, on lines 492-494: "The two intervals before and after this shift are indicative of a change in major dust controls. For the period prior to 6100 yr BP, dust input is reflective of more local controls, whilst the most recent 6100 yr BP of deposition may be linked to more distal forcings."

Comments on Figures Figure 3: any influence of density changes in the measured counts? see note for figure 4.

R23: As mentioned above, we have normalised all ITRAX data (relative to scattering) to take into account density shifts.

Figure 4: Do you calculate ash content after $950°$ calcination? It could also be a good

indicator for mineral matter in your peat.

R24: Unfortunately, we did not measure LOI after 950°, as we were primarily interested in organic matter, not CaCo3 content.

Since ITRAX was performed on the fresh core section how do you take into account the density and/or porosity changes in the measured counts? I suggest to discuss this point in the ms. What could be the influence of the average 20% variation of the bulk density on the elemental counts? How could you normalize each ITRAX elemental profile?

R25: As outlined earlier, in order to circumvent these issues, we have normalised to the scattering values of the ITRAX, after Kylander et al., 2011.

What is the correlation between ITRAX counts and ICP measurements?

R26: The correlation has been calculated, and is presented in lines 203-206: "To facilitate comparison, we bring both records on the same timescale using a Gaussian interpolation with 100 year time steps and a 300 year window. Pearson's r=0.2649, with a p-value of <0.001, indicative of a significant correlation (See SI 4)"

Figure 5: I would suggest to improve this figure. Add the dust events as grey bars as identified in the two ms of Allan et al. 2013 and Le Roux et al. 2012. Some volcanic events are reported in EGR record: they have to be indicated.

R27: In our opinion adding the dust events as indicated by the other two manuscripts clutters the figure. The data is still presented, and so it is hopefully still clear when those studies indicate increased dust. The volcanic events have been added.

Figure 6: add arrows on the vertical scale trend to indicate wetter or dryer local conditions as identified by the reconstructed water table.

R28: These have been added to all figures displaying the TA record.

Table 2: ? add digits

R29: All issues relating to digits here have been resolved

[Figure]

[Figure]

**Fig. 1.** Updated Figure 3

**Fig. 2.** Updated Figure 4

[Figure]

**Fig. 3.** Updated Figure 5

---

## Author Response (AR1)

The manuscript presents profiles of major elements and paleoecological indicators from an ombrotrophic peat bog in Romania. The paleoclimate records are associated to mineral dust and the discussion is focused around possible interpretations that try to disentangle local from distal signals. The authors interpret the history of the site in function of a superposition of changes on diverse spatial scales, from that of local hydrology to the large scale / hemispheric patterns derived from Greenland ice cores or North Atlantic marine sediment records. The topic is of interest to the paleoclimate community. The work appears well structured and its presentation in the manuscript is generally clear. I think that a better discussion of uncertainties would improve the manuscript.

**We thank the reviewer for their kind words, and constructive review.**

**General comment**

My comment about improving the discussion of uncertainties is articulated in two parts, that have to do with (1) the quantification of the dust flux and (2) the attribution of potential sources.

(1) For the purpose of estimating the dust flux, only the concentrations of Ti from ICPOES are used, although a semi-quantitative comparison is carried out against three major elements counts from XRF. Other studies trying to estimate dust from peat bog records discussed the uncertainties related to this issue (e.g. Marx et al., 2009; Kylander et al., 2016). Please discuss more in detail these aspects, and if possible provide some estimates of the uncertainties.

This is a good point, and I order to address the reviewer's comments, we have included a summary outlining the uncertainties within the updated manuscript, lines 209-219:

"It must be noted here that using Ti alone in dust flux calculations does not allow for reconstruction of all minerals related to dust deposition. Ti, which is lithogenic and conservative, is a major component in soil dust, particularly within clay minerals (Shotyk et al., 2002), but may not be associated with other dust-forming minerals, including phosphates, plagioclase and silicates (Kylander et al., 2016), although our records of K and Si may help indicate changes in deposition rates of these minerals (See Mayewski and Maasch, 2006). As a result, we are unable to infer specific mineral-related changes in the composition of dust. However, Ti alone will record changes in the intensity of deposition of the main dust-forming minerals (Sharifi et al., 2015; Shotyk et al., 2002), and variations in K and Si (particularly with local K- and Si-rich dacites a possible dust source) may further indicate the influx of minerals which are not associated with Ti. Such an approach has been applied successfully to studies of changing dust influx (e.g. Allan et al., 2013; Sapkota et al., 2007; Sharifi et al., 2015), with each study able to identify periods of high and low dust deposition from Tiderived dust flux alone."

(2) The attribution of potential sources is based on a simple analysis of correlations between major elements counts, as discussed by the authors, and is also supported by the interpretation of the evidence from indicators such as testate amoeba and pollen. Nonetheless, it would only be by looking at dust size distributions at the same time, that one could derive more firm conclusions about distal versus local contributions to the dust budget (e.g. Mahowald et al., 2014). If possible, include this kind of data, otherwise please discuss this aspect in the text.

We attempted to address this point via grain size analysis on a Malvern Mastersizer 2000. Unfortunately, due to limited sample material, and the very high organic matter content, obscuration values on the laser particle counter were too low for reliable results to be presented. As Kylander et al. (2016) explain, this is a recurring issue with dust reconstruction within peat cores.

The manuscript has been updated to reflect these attempts between lines 164-169:

**2.10 Grain Size**

"In an effort to indicate distal versus local inputs to the bog via the dust particle size, grain size analysis was attempted using a Malvern Mastersizer 2000. Unfortunately, as also observed in previous studies (Kylander et al., 2016) due to the lack of available sample material, and low minerogenic matter (and correspondingly high organic matter) present in the samples, satisfactory obscuration values were not achieved for most analyses."

**Specific comments**

p. 3, 103-106. Please describe how the cores were packed and stored in the phase between recovery and analyses.

**Adjustment made to lines 105-107:**

"Upon recovery, the material was wrapped in clingfilm, transported to the laboratory, described, imaged, and subjected to further analyses. The core was stored at 3°C"

p. 4, 120-121. Later in the text (line 179) you also mention different detection limits for the different elements. Please provide all this kind of information in the same place in the text, and try to mark it visually in the plots.

The value of 50cps is not a detection limit, it is merely guidance for the interpretation of the results. Indeed, studies have been presented displaying variations not related to noise in Mg analysis, where typical cps values are as low as 5-10 (Dulski et al., 2015). With our data it is clear some fluctuations below 50cps are not just noise, as they are echoed in all three elements, and even within the ICP-OES analysis.

Regardless, since we have now normalised the data with respect to the scattering effect of organic material, this sentence no longer applies to our data and has been removed.

Figures 3 & 4. I think a slight confusion can arise because of the way some of the records are organized between these two figures. For instance, it would be useful to see in Figure 3 the Ti concentration profile from ICP-OES along with the major elements counts from XRF, rather than directly the dust flux which weights in the sedimentation rate and bulk density profiles. On the other hand it would be nice to have all the factors defining the dust profile in the same figure, i.e. Ti concentration profile from ICP-OES, sedimentation rate and bulk density profiles.

*Figure 3 has been updated, with all the factors defining the dust flux; raw ICP-OES values, density and sedimentation rate, displayed on one figure. As such, density has been removed from Figure 4.*

p. 4, 126. What is the depth span of each sample? i.e. was the full core analyzes, or just portions of it?

*Lines* 127-128 *have been updated to indicate each sample was* 1*cm*3 *and that the entire core was analysed:*

*"ICP-OES analysis was carried out on 105 samples of 1cm3 of sediment, roughly every 10 cm, through the entire core."*

p. 5, 160. Harmonize with what you say at line 118. p. 5, 175. What do you mean by significant? Did you apply some statistical test? Otherwise perhaps change with "visible".

*The designation "significant" has now been removed. Additionally, we now clarify the reasoning behind the denotation of dust events in lines 183-187:*

"Such zones are identified as an increase in two or more of the elements above the background deposition (K > 0.001, Si > 0.001 and Ti > 0.004, see dashed line on figure 3). These intervals are further discussed as reflecting major dust deposition events, and are referenced in the remainder of the text using the denotation D01-D10 (Fig. 3). Two exceptions, at the base of the core, close to the transition from lake to bog, and the last 1000 years, due to high noise, are not highlighted."

p. 6, 190. Please discuss the uncertainties in estimating the dust flux.

*These have been outlined earlier in the response, and the discussion has been added to the text (lines 209-219)*

p. 6, 208-211. It is not clear at this point what is the contribution of this kind of analysis to the work.

*Since we do not use the magnetic susceptibility values within our interpretation, these data have been removed from the manuscript, and from figure 4.*

p. 7, 238. The dust "Dn" events are selected based on the occurrences of at least one of the elements form the XRF scan, so it is a bit weird to go through the text until this point with some apparent inconsistency between the discussion of peaks D0 to D3, which are not evident and sometimes in anti-phase the what you call dust record. Please either change your definition of dust event (D) or harmonize the text.

In light of comments from reviewer 2, we have removed D0 from our interpretation of dust events, since it may be simply related to minerotrophy. For events D1-D3, as outlined, we have identified them via one of the elements increasing. We agree some confusion may arise with the interpretation of the 'dust record', which is generally based on the Ti-derived dust flux. As such, we have amended the manuscript on lines 366 & 408, replacing "dust record", with "Ti-derived dust record"

When we are considering the results of more than simply the dust flux, we retain the use of the term "dust record".

p. 7, 244-245. As you discuss below, there is not a correlation, so please rephrase with something like "we compare the timing of the identified dust prepositional events with periods ..."

Adjustment made to lines 265-266:

"Firstly, it is noteworthy that five of the identified dust depositional events may be compared to periods of Rapid Climate Change (RCC)"

p. 7, 260. "Appear to indicate": maybe it would be better "are consistent with", in relation to my comment about the missing information on dust particle size distributions. Please include this kind of data if possible, otherwise add a discussion paragraph about this issue.

"Appear to indicate" has been replaced with "are consistent with" on line 284.

**As outlined earlier in this response, and in the manuscript lines 166-169, particle size variations could not be obtained from this highly organic record.**

Figure 5. Check the units of the upper curves (the two cores from McGee et al., 2013): I think you reported the values with the wrong units, which I think are g/cm2/kyr in the original publication, so there is a factor ten difference. In fact, on this scale you have the same dust flux, if not lower, in the North African plume and Belgium or Romania, which would be weird.

This has been checked, and the reviewer is correct-Figure 5 has been revised accordingly

p. 8, 266-268. It is not very clear what is the relation between these two studies, please rephrase.

*To clarify these studies, their relationship, and locations, the manuscript has been adapted in lines 289-292:*

"Saharan dust in Atlantic marine cores strongly increased at this time, with a 140% rise roughly at 5500 cal yr BP observed on the western Saharan margin (Adkins et al., 2006) with another study indicating a rise by a factor of 5 by 4900 cal yr BP in a selection of similarly-located sites (McGee et al., 2013)."

p. 8, 279-289. Again, particle size distributions would help clarify these issues.

We tend to agree particle size data would help the interpretation, but unfortunately we have been unable to attain it.

p. 8, 291-298. Earlier in the text you mentioned the different mobility of these major elements, and how the similarity of their profiles supports the ombrotrophic nature of the peat bog. Please clarify how this is coherent with your analysis here, which is instead based on the difference between the same elements.

These comparisons are not made with regards to any discussion on ombrotrophy. We maintain the similarity of the records, as evidenced by the strong correlations is indicative of the ombrotrophy of the site. However, since the elemental records are not perfectly correlated, and there are periods within which one of the elements is more enriched relative to others, we believe this approach is valid for distinguishing the geochemical signature of the events.

p. 9, 310-313. It would be interesting to compare Type 2 events with the background signature of non-D periods.

This is an interesting idea, and we have implemented it, with results indicating background peat deposition is similar in makeup to Type 2 events- further hinting at the local nature of such deposition. The manuscript has been updated to include these data in lines 339-342:

"The local nature of such deposition is emphasised by the similarity the depositional signal to background values; the elemental composition outside of dust events. For all data points not considered to be related to dust (or the minerotrophic lowermost section), the Ti-K regression is low (r2= 0.1513) with a gradient of 0.0863."

p. 9, 314. "Fig. 8" should probably be "Fig. 6", please check.

The reviewer is correct; this has been corrected on line 343

p. 12, 429. Was the data interpolated somehow before performing wavelet analysis? What is the pace / temporal resolution of the time series fed to the wavelet analysis software?

Before performing wavelet analysis, we used Gaussian interpolation with a timestep of 4 years and a window of 12 years.

**This is included in lines 161-163:**

"For this analysis, the lithogenic normalised elemental data from ITRAX measurements (Ti, K, and Si) was interpolated to equal time steps of four years using a Gaussian window of 12 years."

**Review 2**
The ms presents a very interesting and complete dataset on Holocene dust reconstruction from East European peat record. The ms is well organized with clear aims. The text is too the point. The approach is quite innovative. See the attached pdf with several remarks and/or suggestions on the text and the figures. For my point of view, two mains points should be completed.

First on the methodological approach I suggest to present the calibration between ITRAX and ICP-OES data to better constrain the proposed approach. I will also discuss the potential influence of porosity/humidity changes in the measured peat section on the number of ITRAX counts. Indeed elemental ITRAX data are often normalized, for instance by the number of measured counts in order to take into changes in peat density/porosity but also on surface irregularities. I would explain why this approach is not useful.

**We thank Dr. Fagel for the comprehensive review of our work. We fully agree, and have addressed both comments in the revised manuscript. Firstly, we now present the correlation, and significance of said correlation, between ICP-OES and ITRAX.**

Secondly, we have normalised our ITRAX data (by incoherent + coherent scattering) with respect to the scattering effect of density, porosity and organic matter changes. As such, all data is now presented as normalised counts per second, and all corresponding further analysis has been performed on such data. Please see the comments below for exact location of added text, and adapted figures.

Second I would complete the text about the relationships between the identified dust events, the wet/dry, cold/warm conditions and the related figures.

We have also addressed these issues; see below for text additions, and figure adaptations

Hoping the review will be helpful to improve this interesting manuscript. Sincerely, Nathalie Fagel Please also note the supplement to this comment: http://www.clim-past-discuss.net/cp-2017-6/cp-2017-6-RC2-supplement.pdf

Specific comments from attached Supplement:

Line 26 unclear older or younger than 6,1 kyr?

"Since" 6100 yr BP; manuscript has been adapted to reflect this

Lines 125-126 For my pint of view those data must be more exploited in the ms. Why do you not present the calibration between ITRAX and ICP data?

This is a valid point and we have performed such measurements. Due to the differences in resolution of the data sets, we have performed a Gaussian interpolation at 100 yr steps, with a window of 300 years in order to put both records on the same timescales. The correlation (Pearson's R, n=105) of this is 0.2649, with a p-value of  $1.416 \times 10^{-8}$ , indicating a significant relationship between the two methods. The apparent weakness of the correlation is likely a function of the early half of the record, where low values are observed in both methods, and so the correlation is simply noise, and so we do not expect a perfect correlation. We have attached a graph to display this here:

Such correlation data has also been added to the manuscript on lines 203-206:

"To facilitate comparison, we bring both records on the same timescale using a Gaussian interpolation with 100 year time steps and a 300 year window. Pearson's r=0.2649, with a p-value of

(2) What is the age uncertainty for each interval according to your age model? It is important to precise since some events cover less than 200 years.

The age model uncertainties range from approximately 20 years at the uppermost sections of the core, to around 150 years by the base of the core (See Fig. 2 and Table 1). As such, events lasting 200 years should be discernible at this resolution.

This has been added to the manuscript on lines 175-176: "Age model uncertainties range from 20 years in the uppermost sections to 150 years at the base of the core."

(3) On figure 3, I would suggest to add a secondary scale for dust flux in order to check for smaller peaks.

We have added such a figure to the supplementary information- see SI 3.

Line 192 You must give more arguments to support this assumption. Do you calculate the correlation coefficient between the two datasets? I would add the figure, even as supplementary data. See also notes on fig. 3 and 4.

Such data has been added to lines 203-206. See above for our description of the process, and a simple graph of said comparison (also added to the manuscript as SI 4).

Line 204 add a synthetic lithological column on Fig; 4 to indicate the transition

See SI 2 for lithological column

Line 225 text unclear.

Text has been altered on lines 246-247:

**"These persist only for the period in which each element is enriched, with such cycles particularly evident within the last 6000 years."**

Line 230 Text unclear. Indeed the similar profiles in the three elements emphasize that there is no post-depositional elemental mobility: all elements behave as conservative in the studied peat. I would reorganize the paragraph.

We have inserted the following into line 254:

" indicating the conservative behaviour of such elements in the studied peat."

Lines 247-248 For your record I would suggest to describe deeper the relationships between air temperature and/or humidity and dust flux. In the Misten dust record most dust peaks coincide with cold events.

We agree such a discussion improves the manuscript, and so we have adapted lines 268-276 to reflect this:

"For example D8, between 3450-2800 cal yr BP falls Europe-wide cold period (Wanner et al., 2011). Such cold-related dust deposition has been observed previously in western Europe. However, within Mohos such a compassion may not be drawn for the majority of dust events. For example, event D9 (860-650 cal yr BP) occurs during the Medieval Climate Anomaly, a period of generally higher European temperatures (Mann et al., 2009) but also one of intense human impact on the environment through deforestation and agriculture (Arnaud et al., 2016; Kaplan et al., 2009). Furthermore, such events within the Misten record (Allan et al., 2013) were also linked to low humidity, whereas the Mohos TA (Fig. 4) record indicates locally wet conditions. This suggests that dust depositional events in this region are a result of a complex interplay of environmental conditions in the dust source areas, rather than simply reflecting locally warm or cold, or even wet or dry periods."

Lines 260-261 also observed in other bog records (misten, EGR)

We have revised the text on lines 284-285 to reflect this:

"a major shift in the controls of dust production and deposition at this time, a shift observed in peatderived dust records from Western Europe (Allan et al., 2013; Le Roux et al., 2012)."

Line 299 it could reflect the minerotrophic stage of the peat, with no meaining on dust flux.

*Type 1 deposition only occurs within D10- the most recent dust event, not D0, the event close to the minerotrophic stage of peat.*

We agree however, that D0 may therefore not be a dust event, but related to minerotrophy, and the transition to ombrotrophy, and so we now do not discuss it as a dust event.

Line 344 not clear with double (.

Text adapted line 372: "(Gradient of samples pre-10,500 yr BP = 0.7429, D10= 1.0637)."

Lines 353-356 D10 is older or very close to the minerotrophic/ombrotrophic transition. I would not give to much interpretation in term of dust flux for this first D event. The peat at that time may be affected by surface runoff.

As mentioned above, we agree with the reviewer's interpretation regarding D0, and so we do not discuss it as a dust event.

Line 373 you must first remove signal from volcanic eruption observed in EGR record (see Le Roux et al., 2012).

Text has been adapted to reflect this in lines 401-403:

"with a large dust flux peaks identified in Switzerland between 9000-8400 cal yr BP with other volcanic eruption-related peaks (See Fig. 5), when there is little evidence of dust input into Mohos."

Lines 414 and 418 Fig .7 - But there are still several meter variations in water table: it is significant, no? Moreover most D events during this interval coincide with shallower water table. Such wetter conditions may limit local erosion and favor distal input. We observe the same relationship for the Misten bog (see Allan et al., 2013).

The DWT values are in cm, something which has now been indicated on the figures. As such, the variations in this period only indicate changes of roughly 2-5 cm change at most. The second point regarding such wet conditions favouring distal deposition is valid, however, and so we have adapted the text accordingly on lines 448-449:

"Such wetter conditions also limit local drought-related erosion, and so may be further evidence of distal dust input at this time (Allan et al., 2013)."

Lines 427-428 you may add that the local input erase any distal input in that case.

The text has been adapted lines 458-459:

*"local dust mobilisation, with K-rich dust present at this time, with local input potentially erasing some distal signals."*

Line 436 The time interval is quite short to evidence cyclcity > 1 kyr. i would suggest to keep "millenial cyclicity".

**Text has been adapted in lines 467 and 470 to reflect this**

Lines 460-463 I would suggest to rewrite point 2. Here it overlaps with the point 3. I would say that two intervals are identified wit different major dust control: 1) dust input over the older interval reflects more local conditions; 2) distal input for the younger interval.

In order to improve the clarity, we have re-written this point, on lines 492-494:

"The two intervals before and after this shift are indicative of a change in major dust controls. For the period prior to 6100 yr BP, dust input is reflective of more local controls, whilst the most recent 6100 yr BP of deposition may be linked to more distal forcings."

**Comments on Figures**

Figure 3: any influence of density changes in the measured counts? see note for figure 4.

As mentioned above, we have normalised all ITRAX data (relative to scattering) to take into account density shifts.

Figure 4: Do you calculate ash content after 950° calcination? It could also be a good indicator for mineral matter in your peat.

Unfortunately, we did not measure LOI after 950°, as we were primarily interested in organic matter, not CaCo3 content.

Since ITRAX was performed on the fresh core section how do you take into account the density and/or porosity changes in the measured counts? I suggest to discuss this point in the ms. What could be the influence of the average 20% variation of the bulk density on the elemental counts? How could you normalize each ITRAX elemental profile?

As outlined earlier, in order to circumvent these issues, we have normalised to the scattering values of the ITRAX, after Kylander et al., 2011.

What is the correlation between ITRAX counts and ICP measurements?

The correlation has been calculated, and is presented in lines 203-206:

"To facilitate comparison, we bring both records on the same timescale using a Gaussian interpolation with 100 year time steps and a 300 year window. Pearson's r=0.2649, with a p-value of <0.001, indicative of a significant correlation (See SI 4)"

Figure 5: I would suggest to improve this figure. Add the dust events as grey bars as identified in the two ms of Allan et al. 2013 and Le Roux et al. 2012. Some volcanic events are reported in EGR record: they have to be indicated.

In our opinion adding the dust events as indicated by the other two manuscripts clutters the figure. The data is still presented, and so it is hopefully still clear when those studies indicate increased dust. The volcanic events have been added.

Figure 6: add arrows on the vertical scale trend to indicate wetter or dryer local conditions as identified by the reconstructed water table.

These have been added to all figures displaying the TA record.

Table 2: ? add digitsAll issues relating to digits here have been resolved

Additional Reference

Dulski, P., Brauer, A., Mangili, C., 2015. Combined μ-XRF and Microfacies Techniques for Lake Sediment Analyses, in: Croudace, I.W., Rothwell, R.G. (Eds.), Micro-XRF Studies of Sediment Cores. Springer, Dordrecht, pp. 325–349.

**Periodic input of dust over the Eastern Carpathians during the Holocene linked with Saharan desertification and human impact**

Jack Longman1, Daniel Veres2, Vasile Ersek1, Ulrich Salzmann1, Katalin Hubay3, Marc Bormann4, Volker Wennrich5, Frank Schäbitz4

[revised manuscript text omitted]
 sediments to bog. Secondly, a very brief enrichment between D9, and D10, related the last 1000 years, due to high variation in the scattering values at this time.noise, are not highlighted. The lithogenic, and therefore soil and rock derived Ti and Si have previously been used as proxies for dust input (e.g., Allan et al., 2013; Sharifi et al., 2015), whilst K covaries with Si  $(R^2=0.9945)$  and so controlling factors on their deposition must be nearly the same similar. For these elements, the periods with inferred non-dust deposition are characterised by values approaching the detection limit (150,  $15_{\tau}$ and 40 cps, respectively). A short period of very high values for all elements (10,000, 1300 and 8000 cps, respectively) is observed between 10,800-10,500 cal yr BP (not shown on diagram), reflecting the deposition of clastic sediments within the transition from lake to bog at the onset of the Holocene. Zones of elevated values (D1-D5), with average cps values roughly Ti=300,  $Si=30_7$  and K=100 and persisting for several centuries each, occur sporadically throughout the next 6000 years of the record, between 9500-9200, 8400-8100, 7720-7250, 6350-59007 and 5450-5050 cal yr BP (Fig. 3). Similarly long periods, but with much higher element counts (Ti=800, Si=60 and K=200 cps) occur between 4130-3770, 3450-2850, and 2000-1450 cal yr BP (D6-8). Two final, short (roughly 100-year duration) but relatively large peaks (D9-10) may be seen in the last 1000 years, between 800-620 cal yr BP (with values Ti=300, Si=40 and K=100 cps) and 60 cal yr BP to present (Ti=300, Si=80 and K=400 cps, respectively).

**3.2.2 Dust Flux**

Using the quantitative ICP-OES values of Ti (in ppm), and equation 1, the dust flux can be calculated (Fig. 3). The ICP-OES Ti record shows very good correlation with the Ti data derived through ITRAX analysis. After To facilitate comparison, we bring both records on the same timescale using a Gaussian interpolation atwith 100 year time\_steps using and a 300 year Gaussian window,  $R^2$ . Pearson's r=0.2649, and with a p-value of  $1.416 \times 10^{-8}$ , indicating<0.001, indicative of a significant correlation. (See SI 4). This further implies indicates the reliability of the XRF core scanning method even for such highly organic sediments (as already suggested by Poto et al., 2014), and validates its usage as toolproxy for deriving dust flux (Fig. 3).

It must be noted here that using Ti alone in dust flux calculations may not allow for reconstruction of all forms of dust deposition. Ti, which is lithogenic and conservative, is a major component in soil dust, particularly within elay minerals (Shotyk et al., 2002). On the other hand Ti may not be associated with other dust forming minerals, including phosphates, plagioclase, and silicates (Kylander et al., 2016), and thus, may help indicate changes in deposition rates of particles bearing these elements. As a result of these limitations, at this stage of research we refrain from inferring changes in mineral composition of dust within the core, and the results should be considered only indicative of general dust deposition. Such an approach has been utilised in a number of previous studies (e.g., Allan et al., 2013; Sapkota et al., 2007; Sharifi et al., 2015).does not allow for reconstruction of all minerals related to dust deposition. Ti, which is lithogenic and conservative, is a major component in soil dust, particularly within clay minerals (Shotyk et al., 2002), but may not be associated with other dust-forming minerals, including phosphates, plagioclase and silicates (Kylander et al., 2016), although our records of K and Si may help indicate changes in the composition of dust. However, Ti alone will record changes in the intensity of deposition of the main dust-forming minerals (Shotyk et al., 2002), and variations in K and Si (particularly with local K- and Si-rich dacites a possible dust source) may further indicate the influx

[revised manuscript text omitted]